# FaCT: Faithful Concept Traces for Explaining Neural Network Decisions

**Amin Parchami-Araghi**     **Sukrut Rao**     **Jonas Fischer**[†]     **Bernt Schiele**[†]

`{amin.parchami,sukrut.rao,jonas.fischer,schiele}@mpi-inf.mpg.de`

Max Planck Institute for Informatics, Saarland Informatics Campus, Saarbrücken, Germany

## Abstract

Deep networks have shown remarkable performance across a wide range of tasks, yet getting a global concept-level understanding of how they function remains a key challenge. Many post-hoc concept-based approaches have been introduced to understand their workings, yet they are not always faithful to the model. Further, they make restrictive assumptions on the concepts a model learns, such as class-specificity, small spatial extent, or alignment to human expectations. In this work, we put emphasis on the faithfulness of such concept-based explanations and propose a new model with model-inherent mechanistic concept-explanations. Our concepts are shared across classes and, from any layer, their contribution to the logit and their input-visualization can be faithfully traced. We also leverage foundation models to propose a new concept-consistency metric, $C^2$-score, that can be used to evaluate concept-based methods. Compared to prior work, we show that our concepts are quantitatively more consistent and that users find them to be more interpretable, while retaining competitive ImageNet performance. [1]

## 1 Introduction

Deep learning has proven effective across a wide range of tasks, yet understanding the inner workings of such models remains a challenge, which is critical for their use in sensitive applications such as healthcare. Attribution methods [5, 54, 56] have been typically used to understand the decisions of deep models, but they only show *where* in the input features of importance are located, and not *which* high-level concepts a model uses.

To address this, concept-based explanation methods have become a popular tool to decompose model decisions [9, 35] and arbitrary activations [18, 22] into high-level human-interpretable concepts. These include part-prototype networks [9, 57] and concept bottleneck models [35, 42, 49] that aim to create inherently interpretable models, yet often provide unfaithful explanations [27, 39, 61]. Other approaches like CRAFT [18] decompose model activations *post hoc*, but the extracted concepts are not directly used for prediction, leading to reliance on approximate methods to estimate the importance of concepts to the output [17] and to visualize which region of the input the concept is activated for [52, 61, 62], which may also not be faithful to the original decision-making [2]. Furthermore, concepts are often subject to restrictive assumptions, e.g. being class-specific [18, 22, 36], being limited to small patches or object parts [18, 22, 57], or belonging to a pre-defined set [35, 42], hindering such methods from faithfully explaining the true concepts used by the model.

In this work we propose FaCT, an inherently interpretable model that provides concept decompositions that are *faithful-by-design* (Fig. 1). Our model uses B-cos transforms [8] across its layers to facilitate obtaining faithful attributions for any activation, and sparse autoencoders (SAEs) [6] at intermediate

---

[1]Code available at github.com/m-parchami/FaCT. [†]Denotes equal contribution as advisors.

39th Conference on Neural Information Processing Systems (NeurIPS 2025).

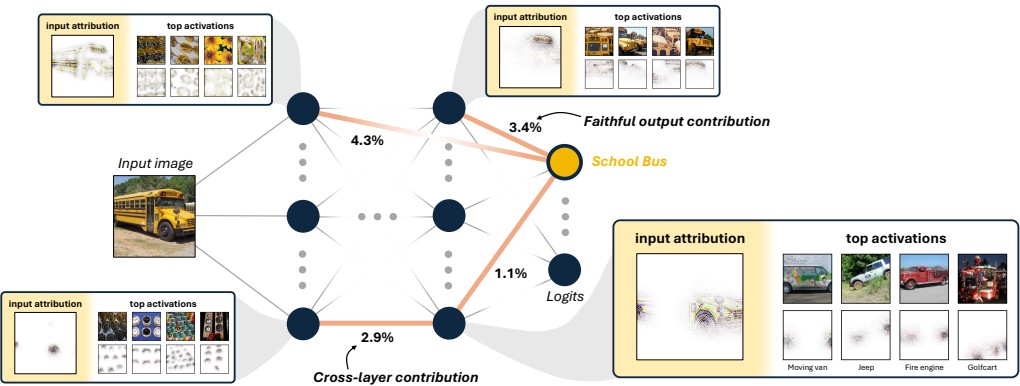

Figure 1: **It All Adds Up**: Our proposed model FaCT offers a faithful concept-decomposition across layers with a shared basis across classes, e.g., the late-layer 'wheel' concept or early-layer 'yellow' concept are shared across classes and used by the model. Further, every concept is *faithfully* visualized at input-level (**Concept Activation = $\sum$ Pixel Contributions**) and every logit is *faithfully* explained at concept-level (**Logit = $\sum$ Concept Contributions**), e.g. yellow-color concept contributes to 4.3% of School Bus logit. Also contributions between different concept layers can be faithfully computed.

layers to decompose the activations into interpretable concepts. The use of a B-cos architecture together with concepts being part of the model's forward pass ensures that (1) model decisions can be faithfully attributed to concepts, and (2) concept activations can be faithfully visualized at input. This is characterized by the concept attributions *adding up* to the logit and pixel attributions *adding up* to the concept activations. FaCT can also decompose across layers to build a concept hierarchy, see e.g. Fig. 1, where the school bus logit can be *fully decomposed* to high-level late-layer concepts as well as simpler early-layer concepts, with each being faithfully visualized. Unlike CRAFT [18] or VCC [36], our concepts are shared across classes (see late-layer 'wheel' concept in Fig. 1), which provides a shared basis that aids misclassification analysis (Fig. 8), and do not include any assumptions on size or spatial extent, leading to a diverse concept decomposition (Fig. 5-right).

Since the concepts used by the model may not align with any predefined human-annotated object parts (see Fig. 3, where annotations fail to capture our concepts), we also find that existing metrics for evaluating concept consistency [30] that make such assumptions are suboptimal. To address this, inspired by its recent success for semantic correspondence [3, 65, 66], we utilize DINOv2 [44] features to evaluate concept consistency without human annotations. Our proposed $C^2$-score takes into account the input features that activate the concept and evaluates consistency independent of pre-defined annotations, while correlating well with human notions of consistency.

Our contributions are thus as follows:

- We propose a model with inherent concept-basis that is used as part of the forward process. The concepts are shared across classes and exist across depth and **generalize across architectures (CNNs and ViTs)**.

- We demonstrate how to **faithfully measure contribution of every concept to the output**, and quantitatively show it outperforms existing approximate concept-importance measures.

- We demonstrate how to **faithfully visualize every concept at input-level** and through a user-study with control groups show how such visualizations impact the interpretability.

- We propose a **novel concept-consistency evaluation metric** for concepts across images for both shared (ours) and class-specific (prior-work) concept sets.

Our models remain competitive on ImageNet [12], while providing faithful concept-based explanations with a diverse (shared) concept basis. We quantitatively demonstrate that our concepts are more consistent, and through a user-study, that they are more interpretable than baselines.

## 2   Related Work

**Concept Extraction Methods** [1, 18, 36, 43] help understand a model's activations in a post-hoc manner by decomposing them into a set of high-level human-interpretable concepts. This decomposition is typically unsupervised, using techniques such as non-negative matrix factorization [10, 18], hierarchical clustering [43], pattern mining [19], or by directly using channels of the layer being examined [1, 15]. However, grounding such concepts in the input requires using post-hoc attributions which may not be faithful [2, 34] to the model. The use of channels as the concept basis [1, 15] also assumes that the model learns separate human interpretable concepts per channel, which need not be true [16, 26]. Further, some approaches [18, 36] use a class-specific concept basis which provides a limited view on how the model shares concepts across classes. In our work, we propose a model with inherent concept-based explanations using B-cos layers [8] that ensures that concepts can be faithfully grounded by design. We use sparse autoencoders (SAEs) to extract concepts at each layer, which provides a shared class-agnostic basis of decomposed concepts.

**B-cos Networks** [7, 8] are a class of inherently interpretable models that use architectural modifications to emphasize weight-input alignment, leading to faithful and interpretable attributions of the model's decisions (cf. Eq. (3)). Since their introduction, B-cos models have proven to extend to large-scale training schemes [4, 20] and their faithful attributions are shown to be a strong proxy for guiding these models [46, 48]. However, B-cos models only provide a local attribution that highlights pixels of importance and do not explain what *concepts* the model uses (Fig. 8). In our work, we design a model with B-cos layers and sparse autoencoder bottlenecks to extract concepts, leveraging the dynamic-linearity of B-cos to obtain faithful and interpretable attributions for grounding concepts, as well as obtaining faithful attributions on how the concepts contribute to the output.

**Sparse Autoencoders (SAEs)** [6, 32, 21, 37, 49, 64] have recently become popular as a tool to decompose model activations into a sparse set of human interpretable concepts, and have been used for understanding large language models [6, 32] and vision models [37, 49, 64] with downstream uses such as model steering [11] and constructing concept bottlenecks [49]. In our work, we propose to use bias-free SAEs to extract concepts across different layers in an image classification model with B-cos layers, and build an inherently interpretable model with a concept hierarchy.

**Inherently Interpretable Concept-based Models** such as Part Prototype Networks [9, 13, 40, 57] and Concept Bottleneck Models (CBMs) [35, 42, 45, 49, 63] first predict a set of concepts using the feature extractor, based on image patch similarity or annotated labels, and then use these concepts for classification. However, the feature extractor is still uninterpretable, and attempts to ground concepts from such models have suggested that they may not be faithful [27, 31, 39, 61]. Recent work [55, 60] explored using B-cos models for better prototypes, however, in contrast to them, we directly use B-cos attributions to obtain fine-grained explanations of concepts.

**Evaluating Explanation Methods** is essential to ensure that explanations can be trusted and are interpretable. For attribution methods, many works proposed sanity checks for measuring their faithfulness [2, 34], metrics for evaluating their localization ability [53, 58], as well as synthetic datasets [25, 50] for a controlled evaluation. For concept-based explanations, prior work [30] proposes evaluating concepts in terms of binarized attributions having high IoU with part annotations. This however is limited to class-specific concepts, requires annotations for every part, and assumes every concept should correspond to an annotated part. In this work, we propose a novel consistency metric $C^2$-score, which leverages foundation models to evaluate consistency of concepts in a class-agnostic manner. Close to our metric are works that measure 'monosemanticity' of individual neurons [16] as the similarity of a set of images or crops. In contrast, our work considers full-image attributions and, through appropriate baselines, can evaluate various concept extraction methods.

## 3   Faithfully Explaining with Concepts

In the following, we describe how our proposed model FaCT provides the user with inherent concept-based explanations (Section 3.1). Since our model leverages B-cos transforms [7] and SAEs [32], we first provide a brief introduction to each respectively. In Section 4 we propose our novel concept consistency evaluation metric $C^2$-score.

**Notation.** Given matrix $\mathbf{M} \in \mathbb{R}^{D_1 \times D_2}$ and vector $v \in \mathbb{R}^{D_2}$: ReLU(.) clamps negative values in a tensor to zero; $c(\mathbf{M}; v) \in \mathbb{R}^{D_1}$ outputs a vector of cosine similarity between each row of matrix

$\mathbf{M}$ with the vector $v$; $(\mathbf{M} \odot v) \in \mathbb{R}^{D_1 \times D_2}$ applies element-wise row scaling $\mathbf{M}$ by $v$; $|v|$ denotes element-wise absolute value; and the $\hat{\cdot}$ operator applies row-wise $\ell_2$ normalization.

**B-cos Networks [7].** For simplicity, let us consider an input vector $x \in \mathbb{R}^K$ and learnable weight matrix $\mathbf{W} \in \mathbb{R}^{H \times K}$ and bias vector $\mathbf{b} \in \mathbb{R}^K$. A conventional ReLU block can be defined as

$$f^{\mathbf{Standard}}(x) = \text{ReLU}(\mathbf{W}\,x + \mathbf{b})\,. \tag{1}$$

A B-cos transform [7] removes bias terms, uses row-normalized weights $\hat{\mathbf{W}}$, and applies cosine non-linearity instead of ReLU. This pushes the rows of $\hat{\mathbf{W}}$ to be aligned with input $x$ for higher activations. The transformation can thus be summarized as a dynamic-linear transform $\tilde{\mathbf{W}}(x)$

$$f^{\mathbf{B\text{-}cos}}(x; \mathrm{B}) = (\hat{\mathbf{W}}\,x) \odot \left| c(\hat{\mathbf{W}}; x) \right|^{\mathrm{B}-1} = \underbrace{(\hat{\mathbf{W}} \odot \left| c(\hat{\mathbf{W}}; x) \right|^{\mathrm{B}-1})}_{\tilde{\mathbf{W}}(x)}\,x = \tilde{\mathbf{W}}(x)\,x\,. \tag{2}$$

A series of B-cos transformations can thus be summarized as a dynamic-linear transform of the input

$$f_{1 \to n}^{\mathbf{B\text{-}cos}}(x) = \left( f_n^{\mathbf{B\text{-}cos}} \circ \cdots \circ f_2^{\mathbf{B\text{-}cos}} \circ f_1^{\mathbf{B\text{-}cos}} \right)(x) = \left( \prod_{i=1}^{n} \tilde{\mathbf{W}}_i(a_{i-1}) \right) x = \tilde{\mathbf{W}}_{1 \to n}(x)x\,, \tag{3}$$

where $a_i$ is the output of layer $i$. Therefore, for every input $x$, B-cos networks can produce model-inherent B-cos explanations $\tilde{\mathbf{W}}_{1 \to n}(x)$ which faithfully reproduce the output logits. For every category $c$, the respective row $c$ of $\tilde{\mathbf{W}}_{1 \to n}(x)$ would serve as the B-cos explanation, i.e. $[\tilde{\mathbf{W}}_{1 \to n}(x)]_c$. As shown in Eq. (3), a series of B-cos transforms can be faithfully summarized as a dynamic-linear combination of input features. In our work, we use this property of B-cos transform to, irrespective of the layer, faithfully compute contribution of each concept to every output logit, as well as, to faithfully visualize every concept at input-level.

**Sparse Autoencoders [32].** In order to discover interpretable concepts from neural network's representations in an unsupervised manner, Sparse Autoencoders (SAEs) [32] have been proposed to obtain a sparse dictionary representation of features. For a given collection of $N$ training features $\mathcal{X} \in \mathbb{R}^{N \times d}$, an SAE is trained to learn a dictionary $\mathbf{V} \in \mathbb{R}^{K \times d}$ and per-feature sparse codes $\mathbf{u} \in \mathbb{R}^{1 \times K}$ as a linear combination of features such that:

$$\forall x \in \mathcal{X} \quad x \approx \breve{x} \quad \text{s.t.} \quad \breve{x} = \mathbf{u}\mathbf{V}, \quad \mathbf{u} = \text{Encoder}(x) = \text{ReLU}(\mathbf{W}x + \mathbf{b})\,. \tag{4}$$

Sparsity can be induced by e.g. $\ell_1$ regularization on the sparse codes, or by explicitly enforcing k-sparse codes, i.e. TopK-SAE [21, 38]. In our work, we use TopK-SAEs, with the encoder having no biases, so the sparse codes can be faithfully represented as a strictly linear combination of features.

### 3.1 FaCT: Faithful Concept Traces

Having B-cos transforms and SAEs introduced, in this section we discuss our proposed model FaCT, and how it explicitly uses a concept-based representation for its decision, while faithfully providing output contributions of every concept, together with faithful input-level attributions from any layer.

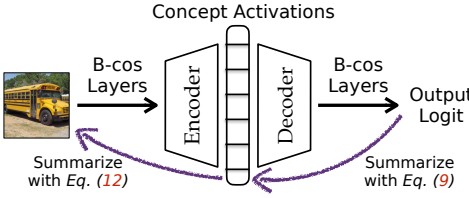

Figure 2: Overview of FaCT.

Let $f_{1 \to n}$ be an $n$-layer deep network of B-cos transforms mapping an input image $I \in \mathbb{R}^{H_0 \times W_0 \times 3}$ to output logits $L^f \in \mathbb{R}^M$ for $M$ categories. We denote intermediate representation $F$ as

$$F \in \mathbb{R}^{H \times W \times C} = f_{1 \to l}(I) \quad \text{s.t.} \quad L^f = f_{1 \to n}(I) = \left( f_{l \to n} \circ f_{1 \to l} \right)(I)\,. \tag{5}$$

We define our SAE module similar to Eq. (4), but without encoder biases. Both encoding and decoding stages are performed independently to all embedding patches, i.e. using $1 \times 1$ convolution operation. We thus transform $F$ to concept-activation tensor $\mathbf{U} \in \mathbb{R}^{H \times W \times K}$ for $K$ concepts

$$F \approx \breve{F} \quad \text{s.t.} \quad \breve{F} = \text{conv}(\mathbf{V}, \mathbf{U}), \quad \mathbf{U} = \text{Encoder}(F) = \text{ReLU}(\text{conv}(\mathbf{W}, F))\,. \tag{6}$$

Our model then uses the reconstructed features $\breve{F}$ for the final output logits $L^{\text{FaCT}}$:

$$L^{\text{FaCT}} := f_{l \to n}(\breve{F}) = f_{l \to n} \circ \text{conv}(\mathbf{V}, \mathbf{U}). \tag{7}$$

Eq. (7) thus demonstrates that the output logits $L^{\text{FaCT}}$ are *only* based on concept activations $\mathbf{U}$. Since $f_{l \to n}$ is composed of B-cos layers, we can summarize it as a dynamic-linear transform, thereby explaining every logit $L_c^{\text{FaCT}}$ for category $c$ as a dynamic-linear combination of concepts $\tilde{\mathbf{W}}(\mathbf{U})$:

$$L_c^{\text{FaCT}} = [f_{l \to n}(\breve{F})]_c = \sum_{i,j,ch}^{H,W,C} [\tilde{\mathbf{W}}_{l \to n}(\breve{F})]_c \breve{F} = \sum_{i,j,ch}^{H,W,C} [\tilde{\mathbf{W}}_{l \to n}(\breve{F})]_c \text{conv}(\mathbf{V}, \mathbf{U}) \tag{8}$$

$$= \sum_{i,j,k}^{H,W,K} \tilde{\mathbf{W}}(\mathbf{U})\mathbf{U} = \sum_{k}^{K} \sum_{i,j}^{H,W} \tilde{\mathbf{W}}(\mathbf{U})_{i,j,k} \mathbf{U}_{i,j,k} = \sum_{k}^{K} \text{Contribution}_k^c, \tag{9}$$

where $\text{Contribution}_k^c$ denotes the contribution of concept $k$ to category $c$. Therefore, the logit is a summation of concept contributions, regardless of the layer, which means FaCT provides model-inherent concept contributions. This is in contrast to prior work [17, 18, 36] which do not directly use concepts for computing the logits and rely on post-hoc measures for concept importance.

Further, since in Eq. (6) we use a bias-free SAE, the entire concept activation tensor $\mathbf{U}$ can be faithfully explained as a dynamic-linear transformation of intermediate features $F$ (see Appendix H on why ReLU is dynamic-linear). As the previous layers $f_{1 \to l}$ leading to $F$ have B-cos transforms, we can faithfully attribute our concept activations $\mathbf{U}$ to input pixels:

$$\text{Concept Activation}_k = \sum_{i,j}^{H,W} \mathbf{U}_{i,j,k} = \sum_{i,j}^{H,W} \left[ \text{ReLU}\big(\text{conv}(\mathbf{W}, F)\big) \right]_{i,j,k} \tag{10}$$

$$= \sum_{i,j,c}^{H,W,C} \left[ \tilde{\mathbf{W}}(F) F \right]_{i,j,c} = \sum_{i,j,c}^{H,W,C} \left[ \tilde{\mathbf{W}}(F) f_{1 \to n}(I) \right]_{i,j,c} \tag{11}$$

$$= \sum_{i,j,c}^{H,W,C} \left[ \tilde{\mathbf{W}}(F)\big(\tilde{\mathbf{W}}_{1 \to l}(I) I\big) \right]_{i,j,c} = \sum_{i,j,c}^{H_0,W_0,3} \left[ \tilde{\mathbf{W}}(I) I \right]_{i,j,c}. \tag{12}$$

We can therefore represent each concept activation as a dynamic-linear combination of input pixels at input-level, which can be visualized similar to the original B-cos [7] explanations. This is in contrast to using approximate visualizations, e.g. showing image crops or up-sampled heatmaps [9, 18, 36, 57].

We have now demonstrated how FaCT faithfully explains every logit as a summation of concept contributions (Eq. (9)) and how it faithfully attributes concept activations to the input (Eq. (12)). In Appendix H, we further derive how one can faithfully visualize $\text{Contribution}_k^c$, i.e. the *output-contribution* of a concept, as well as how one can measure cross-layer concept contribution, e.g. see Fig. 1. The next section focuses on evaluation of concepts, where we propose our novel $C^2$-score that can robustly evaluate consistency of concepts across images in a class-agnostic manner.

## 4   Concept Consistency Evaluation

A systematic evaluation of concept consistency, i.e. whether a concept activates for similar patterns across images, has been of great interest to the community. Prior work [30] proposes to use human-annotated part masks [23]. This however is limited to a small set of classes with class-specific parts and assumes that every concept should correspond to an annotated part. Further, the set of concepts may be shared across multiple (non-object) classes; see how in Fig. 3 the annotations do not match the concepts. To tackle this, we propose $C^2$-score, which uses foundation models' features to measure the concept consistency across images in a class-agnostic manner. We consider a model $\mathcal{H}(I) \in \mathbb{R}^{H \times W \times 3} \to \mathbb{R}^{H \times W \times E}$ transforming image $I$ to a feature-tensor of the same resolution. We use DINOv2 [44] for its success on tasks such as semantic correspondence [3, 65, 66]. We also apply LoftUp [29], an off-the-shelf feature upsampler, to obtain high-resolution feature maps. The LoftUP features are additionally centered using the mean feature computed over the dataset (see

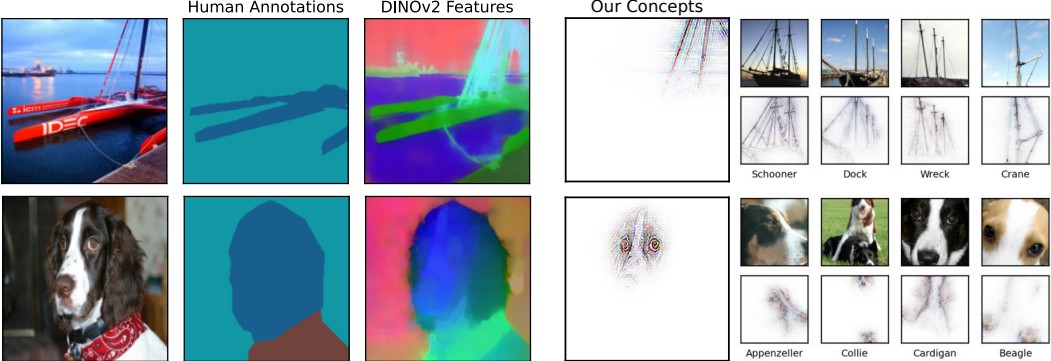

Figure 3: **Beyond Annotations:** We observe that annotations [23] fail to capture our concepts, either by not having them annotated ('ship mast' top-right) or not matching the granularity ('dog blaze' bottom-right). Our proposed $C^2$-score tackles this by considering concept attributions together with DINOv2 features, leading to a class-agnostic evaluation framework. See also Appendix C.1.

Appendix C.3 for further details). For every concept $k$ from the concept set and image $I$ from dataset $\mathcal{D}$, we define the concept activation value $S^{k,I}$ and input attribution $\text{Attr}^{k,I}$ as the following:

$$S^{k,I} \in \mathbb{R}_{\geq 0} \quad \text{Attr}^{k,I} \in \mathbb{R}_{\geq 0}^{H \times W} \qquad \text{s.t.} \qquad \sum_{I \in \mathcal{D}} S^{k,I} = 1 \quad \sum_{i,j}^{H,W} \text{Attr}_{i,j}^{k,I} = 1 \,. \qquad (13)$$

For each concept $k$ activating on image $I$, we can now define the embedding $\mathcal{E}^k(I)$, representing *what* the activation of concept $k$ corresponds to in the output representation of $\mathcal{H}$.

$$\mathcal{E}^k(I) \in \mathbb{R}^E := \sum_{i,j} \mathcal{H}(I)_{i,j} \text{Attr}_{i,j}^{k,I} \,. \qquad (14)$$

We then measure the consistency of such embeddings to each other over the set of images $\mathcal{D}$.

$$\text{Consistency}^k \in \mathbb{R} := \sum_{(I,J) \in \mathcal{D}^2, I \neq J} S^{k,I} S^{k,J} \cos(\mathcal{E}^k(I), \mathcal{E}^k(J)) \qquad (15)$$

Notice that Eq. (15) is defined over the set $\mathcal{D}$. This would be biased towards methods with class-specific concepts, where each concept is only evaluated on the set of images of the same category. To ensure comparability across methods, we define $C^2$-score as the difference between the consistency of the concepts and that of a 'random' concept with a random attribution map on every image.

$$C^2\text{-score} := \frac{1}{K} \sum_K \text{Consistency}^k - \text{Consistency}^{rand} \,. \qquad (16)$$

Therefore, $C^2$-score lends itself as a simple yet robust evaluation framework that takes the concept attribution into account and can evaluate both class-specific and shared concept sets. Having such an evaluation framework, in Section 5.2 we compare our concepts with prior work. In Fig. 4-right, we further validate the proposed $C^2$-score and how it is close to human definition of consistency, by showing *randomly sampled* concepts with high and low $C^2$-score. In Appendix C we elaborate on $C^2$-score, showing how it outperforms annotations and how it correlates with user interpretability.

## 5 Inspecting the Concepts

In this section we inspect the model-inherent concept-based representation of FaCT. We begin by evaluating FaCT across architectures and layers in Section 5.1, measuring ImageNet performance and concept consistency ($C^2$-score), together with concept diversity in terms of spatial extent. Afterwards, Section 5.2 quantitatively compares consistency of our concepts to prior work. Lastly, Section 5.3 compares the interpretability of our concepts and visualizations across layers compared to baselines in a user study. With our concepts evaluated in this section, in Section 6 we focus on the decision-making

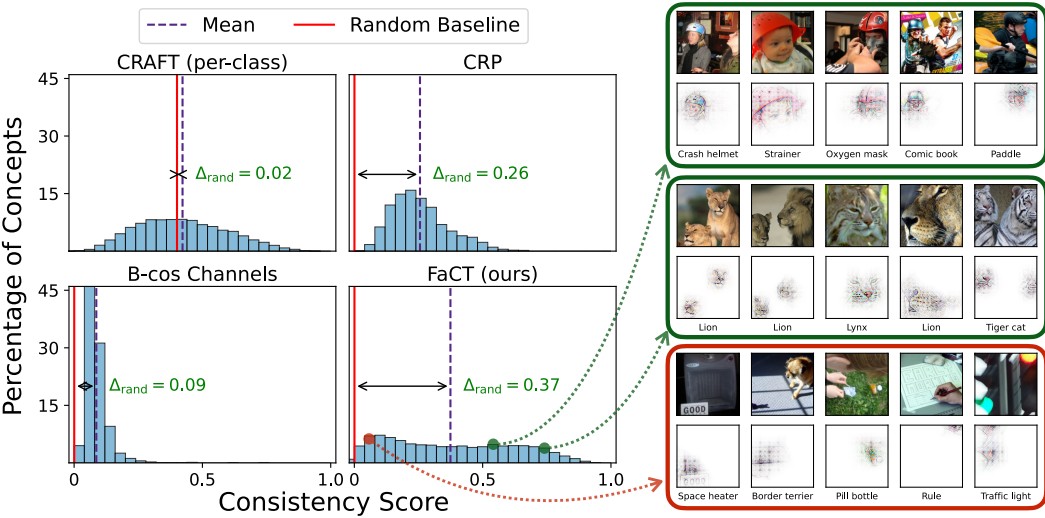

Figure 4: **Evaluating Concept Consistency:** We evaluate the $C^2$-score (cf. Section 4) for both FaCT's concepts and prior work's. **(left)** we plot the percentage of concepts for different consistency ranges, finding our concepts to be more consistent than those of prior work. **(right)** We *randomly* sampled concepts from different ranges of $C^2$-score, to demonstrate the effectiveness of the $C^2$-score. Notice that the $C^2$-score correctly assigns high consistency to the 'helmet' or 'muzzle' concepts, despite them being shared across classes. See also Appendix C.

of the model, by quantitatively evaluating concept contributions (Eq. (9)) to prior works' approximate measures, as well as leveraging the shared concept basis of FaCT to study misclassification.

Before we begin, let us briefly discuss our implementation details. We used ImageNet-pretrained B-cos [8] networks with fixed parameters and used bias-free TopK-SAE [21, 38] sweeping $TopK \in (8, 16, 32)$, with $K \in (8192, 16384)$ total concepts. We constructed the training dataset by sampling feature patches from ImageNet's training set. Throughout the paper, each concept is visualized by top-activating test images, with attribution (Eq. (12)) and the image category. See also Appendix A.

## 5.1 FaCT is Competitive and Diverse

We begin by evaluating FaCT across layers (early, mid, late) and architectures (CNNs [24, 28] and ViTs [14]) on ImageNet [51]. In Fig. 5-left, we observe that our models are able to maintain competitive accuracy compared to original B-cos, despite being trained on having a sparse concept-representation, even at intermediate layers. We observe that for a small drop in performance ($< 3\%$ across), we are able to obtain significantly higher $C^2$-score for our concepts, e.g. 0.11→0.39 for DenseNet at Block 3/4. In fact, in Section 5.3 we will also show users found our concepts more interpretable than the baseline. We also see that FaCT readily generalizes to ViT architectures. Further, in Fig. 5-right, we plot the diversity of our concepts in terms of spatial extent, i.e. number of highest-attribution pixels needed to cover 80% of the total attribution. We observe concepts of variety of sizes across layers and architectures, e.g. see the small late-layer 'Bike Helmet' concept for DenseNet or the large early-layer 'Wooden Texture' for ViT. This is in contrast to prior work [18, 36, 40] that assume fixed sizes for every concept. Further analysis and visualizations can be found in Appendix D.

## 5.2 FaCT is more Consistent

As discussed in Section 3, we leverage SAEs for extracting concepts at intermediate layers, while faithfully visualizing every concept at input-level. In Section 4, we also discussed how our proposed $C^2$-score is a stronger benchmark for evaluating consistency of concepts. In Fig. 4, we evaluate the consistency of our concepts using $C^2$-score, as well as concepts from prior work. In Fig. 4-left, we observe that our concepts are more consistent than prior work, with significant increase compared to B-cos channels ($C^2$-score 0.09→0.37). We also *randomly* sampled concepts from different consistency

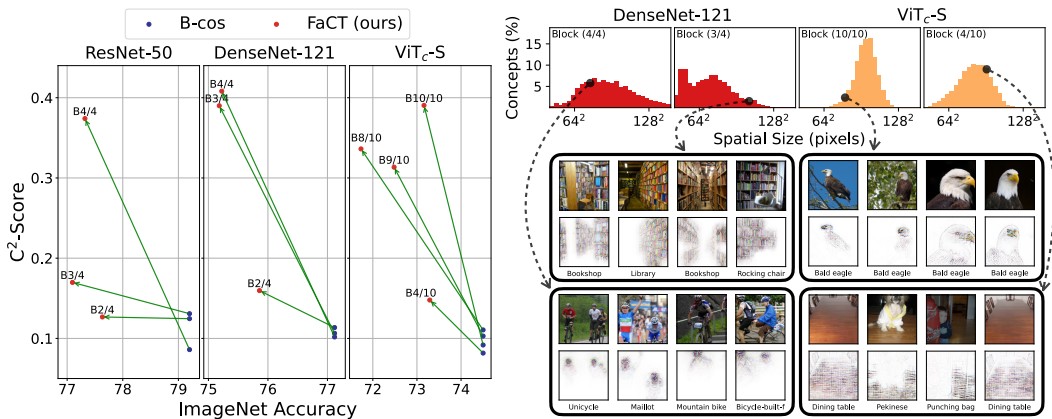

Figure 5: **FaCT for Diverse Concepts. (left):** We observe significant gains in terms of concept-consistency for FaCT compared to B-cos channels. This holds across architectures (columns) and layers (points) with competitive performance on ImageNet (largest drop < 3%), see Appendix B for further analysis and comparison to standard models. **(right):** We observe high diversity for our concepts in terms of spatial extent (top) and show samples at the bottom. See also Appendix D.

ranges and visualized the concepts in Fig. 4-right. We observe consistent concepts such as 'Helmet' and 'Muzzle' obtain higher scores than inconsistent ones. Notice that here the consistent concepts are shared across classes, yet $C^2$-score reliably assigns them high consistency scores.

### 5.3 FaCT is more Interpretable

As explanations should aid humans to gain insights, we evaluate the interpretability of concepts based on whether users can retrieve a meaning from our visualizations. In particular, we randomly sampled 100 early and 100 late concepts from FaCT, together with 30 late and 30 early randomly chosen B-cos channels as a baseline. For each concept, we visualized top-10 activating images along with their input attribution for each image. We randomly assigned participants to one of 10 groups to rate the shown figures in terms of interpretability on a 5-point scale (higher is better). Each group was shown samples from all cases. We further conducted a counterbalanced AB/BA study to evaluate whether our faithful input attribution increases interpretability. For each of our concepts, we generated one version with and one without the input attribution as control. Each group experienced concepts from the experimental and from control condition. The study was conducted fully anonymized with 38 volunteer participants. For further details and sample questions, refer to Appendix E.

The study results in Fig. 6 show that the participants found our concepts far more interpretable than B-cos channels, for both early- and late-layer concepts. We also observe that providing our input attributions can change the given scores, observing that on average they lead to an increase in interpretability, particularly for earlier layer concepts with about 0.5/5 average increase. In Fig. 6-right, we visualize four early-layer concepts with the largest increase in score between their two groups. These are low-level visual patterns such as background (+3.0/5) and up-facing curves (+2.5/5) which users were able to interpret only when aided with our faithful input-level visualizations.

Our results thus find FaCT's concepts to be more consistent (Fig. 4) and interpretable (Fig. 6) than the baselines. This was achieved by solely relying on faithful visualization of concepts without putting any assumptions such as having a fixed spatial-size, being only object parts or class-specific, or coming from a pre-defined concept set, as often seen in prior work [9, 13, 18, 35, 40, 42, 57]. That said, aligning the concepts with textual labels may be of interest to some users. For this, one can accompany FaCT with existing neuron-labeling methods. In particular, we applied CLIP-Dissect [41] to our SAE latents, using common English words to match each concept to the text label with the most similar activation statistics in CLIP [47]. This results in concepts (A-F) in Fig. 8 to be named as follows (best of top-three per concept): A→'balls'; B→'jerseys'; C→'rugby', D→'flexible'; E→'volleyball'; and F→'basketball'. We thus observe that FaCT's concepts also lend themselves well to be named using existing neuron labeling methods. In the next section, we turn our attention towards contribution of concepts to the predictions.

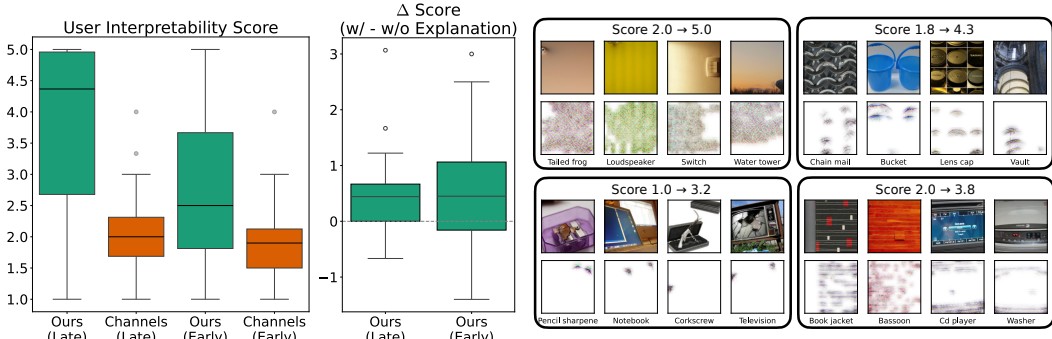

Figure 6: **User Study on Concept Interpretability: (left)** We evaluate the interpretability and consistency of concepts and visualizations through a user-study. We observe significant gains in terms of interpretability compared to B-cos channels. **(center)** We show the results of our control study for concepts with average score $\leq 2.5$ when viewed only with images, and observe that the explanations increase their interpretability score, in particular for earlier layers. **(right)** we show four early-layer concepts, which had the highest score increase when users saw the explanations, e.g. for the 'curve' concept (top-right), the explanations increased the score by $\Delta = 2.5(/5)$. See also Appendix E.

## 6 Inspecting the Decision-Making

As in Section 3.1, our proposed model uses a shared concept basis and is able to faithfully decompose the output logits into concept contributions Eq. (9). We next empirically evaluate our concept-contributions compared to approximate importance measures from prior work [22, 18, 36] and then study a misclassification case to understand the model's confusion on a concept level.

**Validating concept contributions.** We further empirically validate the faithfulness of our concept-contributions from Eq. (9) using the concept-deletion metric [17]. In short, this metric removes concepts from most to least important under different importance measures, and compares the drop in logit. As our concept-basis is shared across classes and the concepts are in fact used to produce the output logits, we additionally evaluate the overall accuracy drop when removing the concepts. This gives a broader understanding of how *all of the outputs* change as the concepts are removed. In Fig. 7 we observe that our concept contributions, as defined in Eq. (9) provide significantly sharper drops, both for top-1 logit as well as overall accuracy, indicating that the reported contributions are a better signal for measuring importance of concepts. We see significantly higher gains compared to other attribution methods such as Saliency or Sobol indices used in prior work [33, 22, 18]. Further details on this evaluation can be found in Appendix F.

Interestingly, for Block (2/4) in Fig. 7, we see a sharp drop of accuracy with deletion of only a few concepts, which indeed highlights the faithfulness of our concept contributions. In Appendix F we further study the concepts which cause such a sharp drop upon deletion and find them to be a small set of concepts that always contribute to every decision. In Appendix F, we further run a similar experiment as in Fig. 7, but preventing such concepts from being removed, and find that our concept contributions still significantly outperform other baselines.

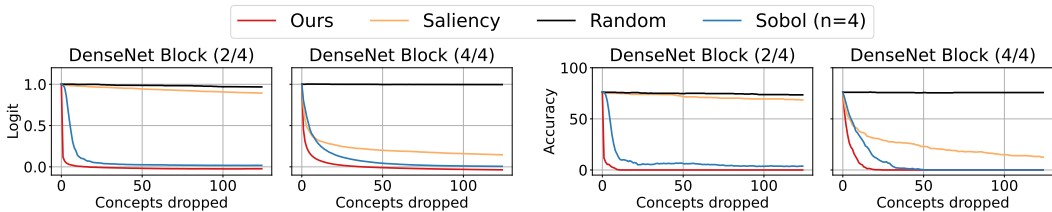

Figure 7: **Concept Deletion:** We iteratively delete concepts in order of importance based on different attribution methods (steeper drop is better). We observe significant improvements using our Eq. (9) compared to existing concept-attribution, especially at earlier layers. See also Appendix F.

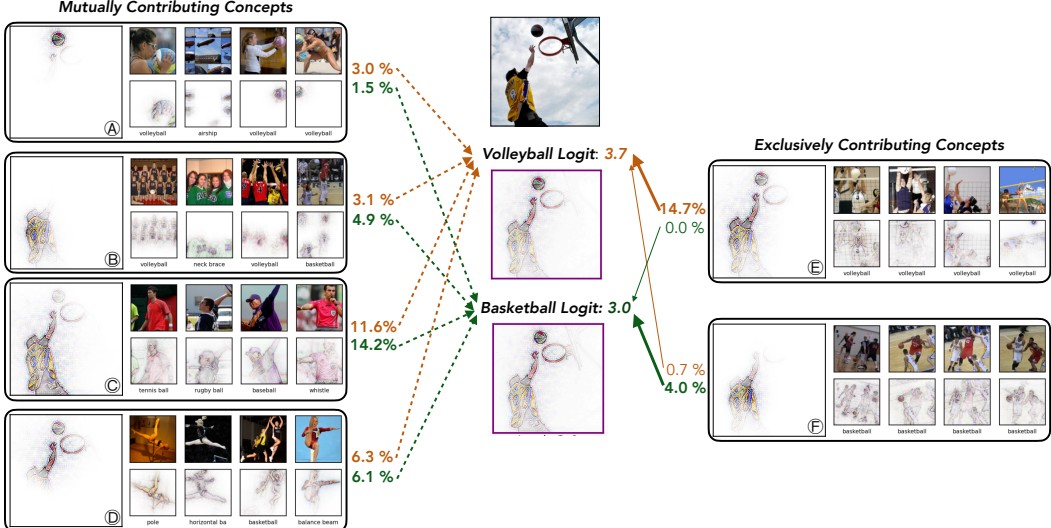

Figure 8: **Understanding What is Shared**: Having a shared concept basis, we inspect a misclassification and attribute each of the logits (i.e. Basketball and Volleyball) to inspect what are the concepts that contribute to the confusion. **(center)**: the two logits with their original B-cos explanation [8] are reported. **(left)**: we see four concepts (A-D) that mutually contribute to both logits. These include common features between the two, such as 'ball' (A) and 'jerseys' (B). **(right)**: we see concepts (E) and (F) that exclusively contribute to volleyball and basketball logits. See also Appendix G.

**Leveraging a shared concept basis.** Through FaCT, we can understand wrong decision-making inside the neural network, such as a Basketball image that is misclassified as Volleyball (Fig. 8). Just by looking at the attribution (in the middle), it is not clear *why* the model is confused. Through our concept-level explanation, we get *which* concepts are active and *what* they perceive (Fig. 8A-F) , as well as *how much* the concept contributes to each logit. This allows to understand the model's confusion on a concept-level. For the mutually contributing concepts, we observe that these are confounding factors, such as 'ball' (A), 'jersey' (B), 'person with shirt' (C), or 'limbs' (D), that appear for both classes. In fact, for the 'jersey' concept (B), we already see samples of both volleyball and basketball classes within the top-activating images. Further details together with comparison to class-specific methods can be found in Appendix G. In summary, FaCT provides us with a deeper understanding where and why model reasoning goes wrong.

## 7  Discussion

In this work we presented FaCT, a model with faithful and inherent concept-based explanations, which uses exactly these concepts for its decision-making. Our model fully decomposes every logit into concept contributions and explains each concept faithfully at input-level, providing interpretable and consistent concepts across layers and architectures while remaining competitive on ImageNet. This was achieved by having the concepts shared across classes and not putting any restrictive assumptions on the concepts. In Appendix D, we further explore the diversity of our concepts, with additional samples across layers and architectures. We also demonstrate the generalization of FaCT to other datasets in Appendix K.

Our work also introduced a concept-consistency metric ($C^2$-score) to evaluate both shared and class-specific concepts at scale, avoiding limited comparison against annotations (Fig. 3), while agreeing with human notion of consistency (Fig. 4). In Appendix C we further demonstrate how $C^2$-score is superior to using human annotations for evaluating concepts.

In summary, we proposed an architecture that inherently encodes *interpretable concepts across its layers*, provides *faithful attribution maps* for each concept exactly reflecting the underlying model perception, and provides *faithful contribution scores* for each concept revealing its exact impact on a downstream classification.

## Acknowledgments

We would like to thank our colleagues Thomas Wimmer, Olaf Dünkel, Moritz Böhle, Nhi Pham, and Siddhartha Gairola for the helpful discussions. We are especially thankful to Thomas for helping with Fig. 1.

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

# FaCT: Faithful Concept Traces for Explaining Neural Network Decisions

## Appendix

In this supplement to our work on faithful concept traces (FaCT), we provide further results, derivations, and implementation details, as indexed below. We **particularly encourage** the reader to see Sec. C where we demonstrate the suitability of our proposed $C^2$-score, as well as, Sec. D where the diversity of our concepts and their generalization across layers and architectures are demonstrated.

# A Training Details

In this section we provide details on our implementation, constructed dataset, training sessions, as well as checkpoint selection.

**Implementation.** As discussed in Section 3.1, our model consists of models with B-cos transforms [S2, S3] with Sparse Autoencoders at layers where we want a concept-based representation. We therefore use pre-trained B-cos checkpoints (i.e. B-cos ResNet-50 [S17], B-cos DenseNet-121 [S18], and B-cos ViT$_c$-S [S8]) from the official release[2]. Within the paper, we refer to the following layer names as such:

- **ResNet-50**: layer2→Block 2/4; layer3→Block 3/4; layer4→Block 4/4
- **DenseNet-121**: transition2→Block2/4; transition3→Block3/4; norm5→Block4/4
- **ViT $_c$-S**: encoder8→Block 8/10; encoder9→Block 9/10; encoder10→Block 10/10

For the SAEs, we base our implementation on the Dictionary-Learning codebase [S27]. We use the TopK-SAE [S14, S26], and modify it to be bias-free (i.e. no data whitening and no biases for encoder or decoder). This is needed so that faithful output contribution in Eq. (9) and faithful input attribution in Eq. (12) can be obtained.

**Dataset.** Our SAEs are trained for reconstruction on ImageNet [S30]'s training set. We first take 50 samples per class as a held-out validation set and leave the rest for training. We accumulate the features from the training set $\mathcal{D}$, but as this would be a very large dataset, especially at earlier layer of CNNs with high spatial dimension. For every image $I \in \mathcal{D}$ we perform importance sampling over the spatial dimension of $F_I \in \mathbb{R}^{H \times W \times E}$, based on the contribution to the output.

$$\mathcal{D}_{\text{SAE}} = \bigcup_{I \in \mathcal{D}} \left\{ F_I[i_n, j_n] \,\middle|\, [i_n, j_n] \sim \text{Contribution}^c_{F_I[i,j]}, \ n = 1, \ldots, M \right\}. \tag{A.1}$$

Thus from every image's intermediate features $F_I$ we randomly sample M feature vectors weighted towards features of higher importance. For convenience, we set the $M$ per model and layer, such that each constructed dataset $\mathcal{D}_{\text{SAE}}$ uses at most 170 GB. The Contribution$^c_{F_I[i,j]}$ is measured within the original B-cos model with B-cos attributions [S3] of the predicted logit to every spatial position in $F_I$, i.e. similar to Eq. (3) but for intermediate layers.

**Training.** For training the SAE, we use a batch size of 32,786 (individual feature vectors), with a sweep of learning rates $\lambda \in [0.001, 0.0001]$, total number of latents $K \in [8192, 16384]$, and sparsity factor of TopK-SAE $topk \in [8, 16, 32]$ (except for ViT @ Block 4/10, where we also tested $topk = 64$, as the lower values led to low accuracy). We use Adam Optimizer [S23], together with cosine learning-rate scheduler, with initial warm up of 2 epochs. We trained each model for 16 epochs, but in general observed that many runs plateaued at earlier epochs.

**Checkpoint Selection.** When training the SAEs, we observed that some runs may end up with many 'dead' latents, i.e. latents that never get activated. We also observed that runs may sometimes have 'always-active' latents, i.e. latents that are active on more than 60% of *the data points* (i.e. features). This can be attributed to (a) the use of TopK-SAE, which may repeatedly take the same subset of latents as the TopK, and (b) the fact that we use a bias-free SAE, as the model may learn a set of 'mean features' that are always active. We observed this also for vanilla SAEs (i.e. with L1 regularization) on our initial experiments and also found traces of this in related work, e.g. see the dots with frequency of 1 in Figure 3 of [S25]. For every sparsity factor, we thus selected top two best performing runs (using held-out set from training data), and for similar performances, chose the one with fewer 'dead' or 'always-active' latents. Finally, we evaluated our set of candidate runs per model and layer on the ImageNet's official test set, which are reported in Sec. B.

The final checkpoints, which were also used throughout the paper for visualizations and evaluations, were obtained with the above procedure. For convenience and reproducibility, we report the configuration of these checkpoints in Table A1.

---

[2]https://github.com/B-cos/B-cos-v2

| Architecture | Position (Layer Name) | Learning Rate | Number of Concepts (K) | Sparsity Factor (TopK) |
|---|---|---|---|---|
| ResNet-50 | Block 4/4 (layer4) | 0.001 | 16384 | 16 |
| | Block 3/4 (layer3) | 0.001 | 16384 | 32 |
| | Block 2/4 (layer2) | 0.001 | 8192 | 32 |
| DenseNet-121 | Block 4/4 (norm5) | 0.001 | 16384 | 16 |
| | Block 3/4 (transition3) | 0.001 | 16384 | 32 |
| | Block 2/4 (transition2) | 0.0001 | 8192 | 32 |
| ViT$_c$-S | Block 10/10 (encoder10) | 0.001 | 8192 | 32 |
| | Block 9/10 (encoder9) | 0.0001 | 16384 | 32 |
| | Block 8/10 (encoder8) | 0.001 | 8192 | 32 |
| | Block 4/10 (encoder4) | 0.0001 | 16384 | 64 |

Table A1: Configuration of chosen checkpoints (based on the procedure described in Sec. A)

# B Additional Analyses on Sparsity-Accuracy Tradeoff

In Fig. B1 we report detailed results on ImageNet. For each architecture, we report standard variant, B-cos, and FaCT for different layers. For standard models, we rely on numbers reported in [S3] which evaluates under comparable settings, except for Standard DenseNet-121, for which there is no checkpoint with modern torchvision recipe available. In Fig. B1 (top-half) we show ImageNet accuracy against per-image $\ell_0$-norm, which shows the total number of concepts that activate per-image for different models (columns) and different layers (colors within each subplot), with the pareto front being displayed as a line, and the selected checkpoint highlighted with star ($\star$).

As one would expect, on the top we see that with higher per-image $\ell_0$-norm we can achieve higher accuracy for our models, and for the same accuracy, one would require higher per-image $\ell_0$-norm for earlier layers. Note however that the total number of concepts per-image are used by all of the logits Eq. (7), and to explain a single logit, one would only require to look at fewer concepts. The bottom half of the plot reports $\ell_0$-norm for covering 80% of the positive contribution per-image.

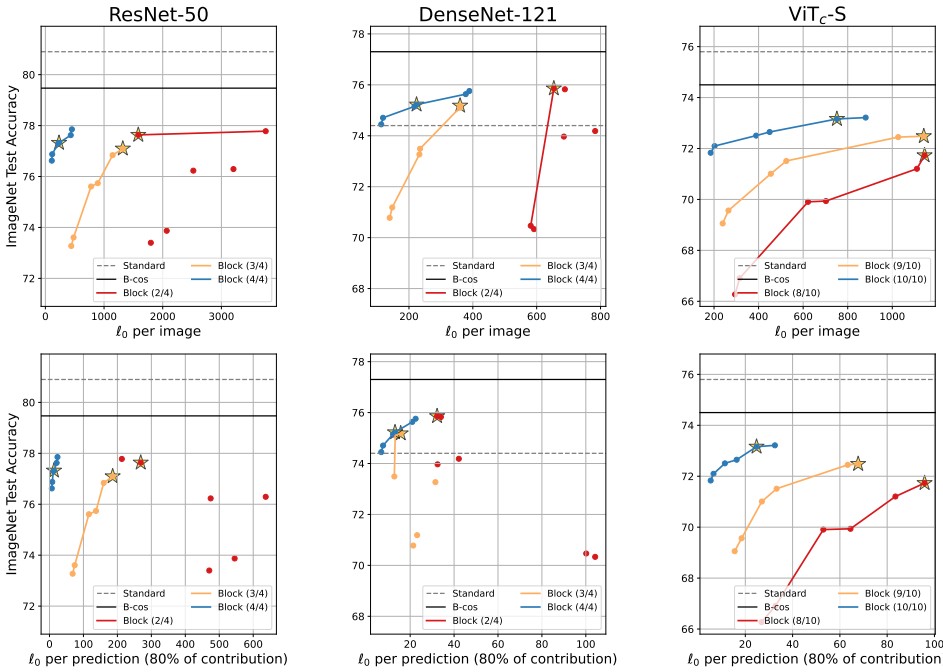

Figure B1: **Complete ImageNet Results with Sparsity-Accuracy Tradeoff. (Top)**: we observe the sparsity-accuracy tradeoff and that with higher per-image $\ell_0$ one can achieve higher accuracy. **(Bottom)**: We observe that for explaining 80% of the positive contribution to the logit, one would need much fewer concepts compared to the above half.

# C    C²-score: Superior to Annotations

In this section we elaborate on our proposed C²-score Eq. (16) for concept consistency. In Sec. C.1, we show limitations of existing human-annotations by showing concepts that do not match annotations. In Sec. C.2 we discuss how C²-score correlates with our user-study results, and lastly in Sec. C.3 we provide further details on the implementation.

## C.1    Comparison to human annotations

In Fig. C1 we show three examples (major rows) from PartImageNet [S16] against the *PCA decomposition* of DINOv2 [S29] features (upsampled with LoftUP [S19]). We find DINOv2 features to be significantly more inclusive than annotations. Note that the DINOv2 features have in fact even richer semantics, as the displayed colors here are simply PCA decomposition of features.

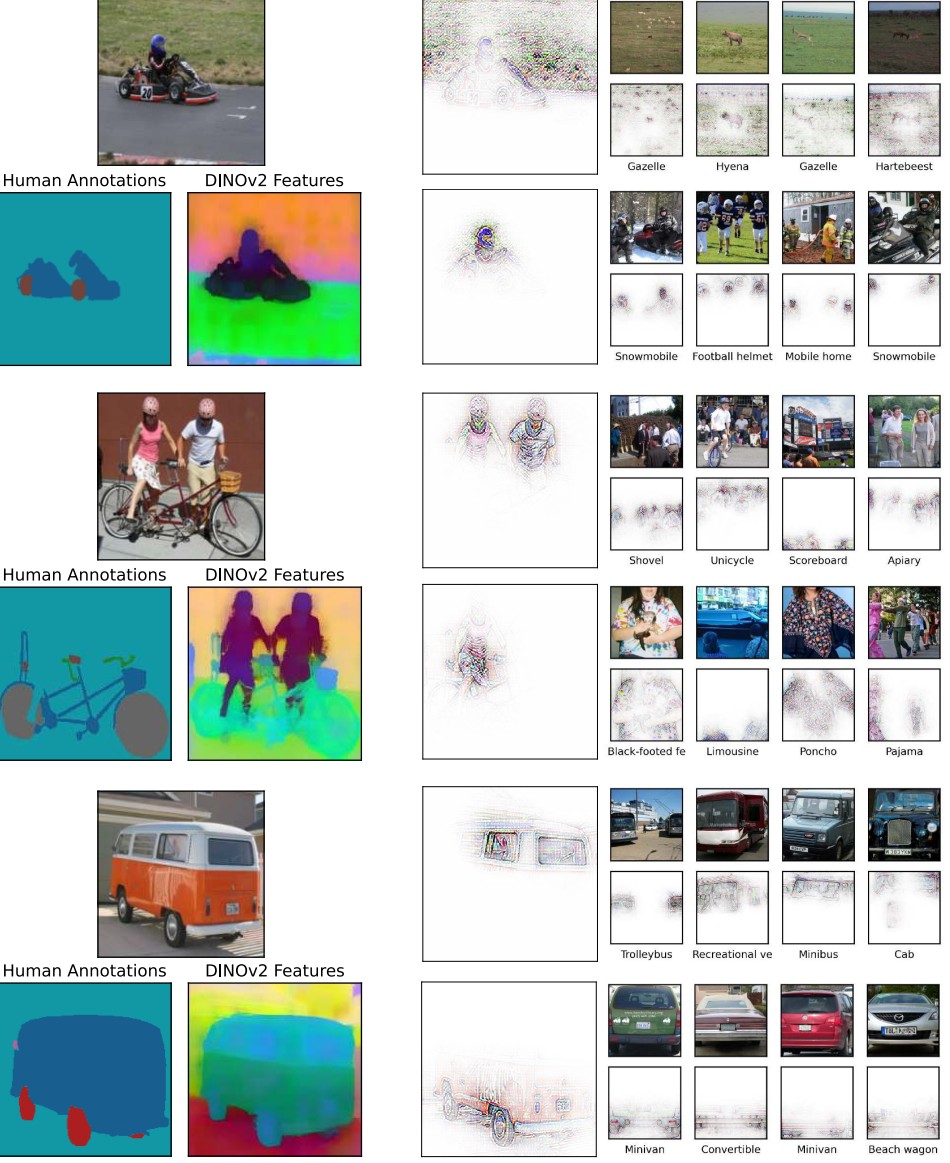

Figure C1: **Comparing DINOv2 features vs. Human Annotations:** We find DINOv2 features, as used in C²-score, to be much superior for evaluating our diverse concepts on the right. For every row, we show two sample concepts from FaCT that are consistently being activated, yet are not considered within the human annotations. On the first and second examples we see ('Grass' and 'Helmet') and ('Person' and 'Floral dress') not being considered within the annotations. Lastly, we see the minivan has concepts such as 'Windows' or 'Chasis' that do not match the granularity of human annotations.

As per Fig. C1, we find using DINOv2 [S29] features under $C^2$-score to be a superior evaluation framework compared to human annotations, which do not cover all concepts and do not support shared concept bases. E.g., concepts such as 'Grass' or 'Helmet' on the top row are not even considered as part of the annotation, which would falsely be indicated as in-consistent IoU under prior metrics [S20]. We however see that, even in the simplified PCA visualization of DINOv2 features, we observe a different color for different parts in the image. Even if a significant part of the image was annotated, e.g. in the last row of Fig. C1 for Mini-van, we again see the granularity of annotations may not match the granularity of concepts. With concepts such as 'Windows' not being considered in the annotation.

## C.2 Comparing $C^2$-score with user study results

To further validate our proposed $C^2$-score, we compared the user interpretability scores from our user study (done for Section 5.3 in the main paper) against our proposed $C^2$-score Section 4. In particular, we compared how the study participants sorted the late-layer concepts that were shown to them (by rating each from 1 to 5), and compare the Spearman correlation of these sorting with the consistency scores that we get per concept, according to Eq. (16) in the paper. Note that, as per Section 5.3, our user study was conducted for evaluation of our concepts compared to the baselines in terms of interpretability. One could further tailor a study solely dedicated to evaluating the $C^2$-score metric itself, e.g. asking users to rate *ranking* of concepts.

Nevertheless, in Fig. C2 we observe that for all of the 38 participants the $C^2$-score's ranking of concept has a positive spearman correlation with how they rated the concepts throughout the study. In particular, 33/38 and 20/38 participants have 'moderate' ($> 0.4$) and 'strong' ($> 0.7$) correlation [S6] respectively. We thus find $C^2$-score lends itself well for evaluating our shared concept-bases in a class-agnostic manner and it correlates well with what users found interpretable, while also acknowledging that a user study dedicated to $C^2$-score could provide further insights.

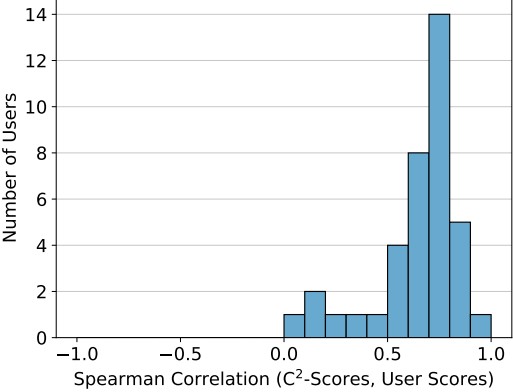

Figure C2: **Comparing User Scores with $C^2$-score for DenseNet-121.** We observe that the rankings achieved by $C^2$-score are highly correlated with how users ranked the late-layer concepts that were shown to them. In particular, all users had positive correlation, with 33/38 having 'moderate' and 20/38 having 'strong' correlation.

## C.3 Implementation Details

As discussed in Section 4, our $C^2$-score leverages DINOv2 [S29] features, in particular DINOv2-Small, together with a state-of-the-art feature up-sampler LoftUP [S19]. We then evaluate the consistency of concepts according to Eq. (16) on ImageNet's official test set.

**Evaluating the Concepts.** For Fig. 4 in the main paper, we evaluated our concepts against baselines on Block 4/4 of ResNet-50 [S17]. For CRAFT [S12] and CRP [S9], we used standard torchvision [S34] checkpoints. We trained CRAFT [S12] with 16 concepts per class for every ImageNet class with 600 training samples and considered the up-sampled NMF-coefficients as the concept-attribution, as done in the original paper [S12]. For CRP [S9], we used the official zennit-based [S1] implementation together with EpsilonFlat rule, as used in the original paper [S1].

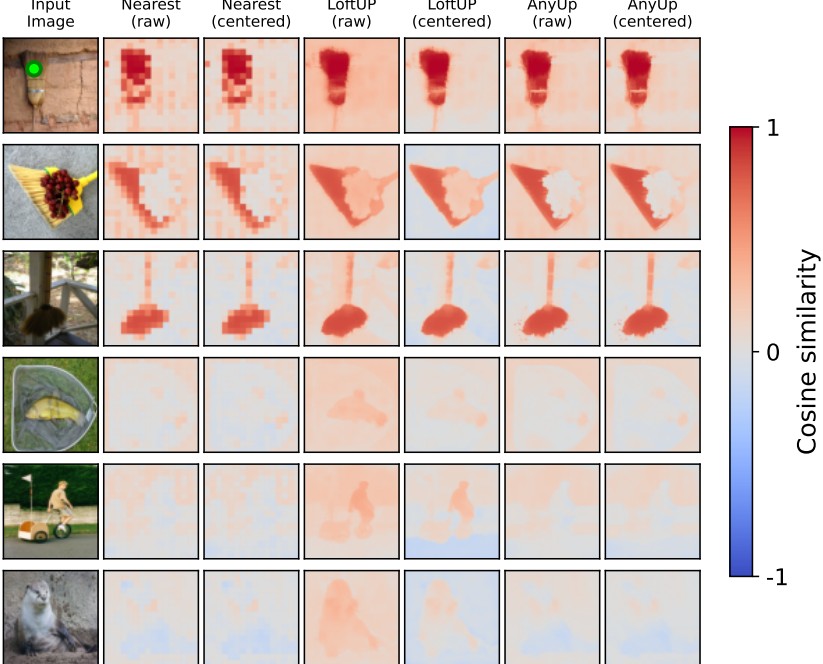

|  | Input Image | Nearest (raw) | Nearest (centered) | LoftUP (raw) | LoftUP (centered) | AnyUp (raw) | AnyUp (centered) |

Figure C3: **Comparing Different Up-samplers.** We measure the cosine similarity of the marked green pixel in the first image (top-left) to all of the other features across different images (rows), under different up-sampling methods (columns). We observe that feature up-samplers (cols. 4-7) provide much sharper feature maps with more accurate details. We observe that LoftUP exhibits a baseline similarity across all feature points, even across unrelated images (bottom-3 rows), which is mitigated when features are centered around dataset-mean. We see that for more-recent up-samplers [S37], the centering is less essential.

For all models, layers, and concept methods in Figs. 4 and 5 in the main paper, we considered top 5% of activating test images for every concept, ensuring to select at least 10 images per concept. For concepts with fewer than 10 activating images, we considered all of their images. We discarded concepts that activate on fewer than 5 images.

**Up-samplers and Centering.** When using LoftUP [S19] (state-of-the-art at the time of submission), we noticed a baseline similarity of up-sampled features, for which we applied dataset-centering, i.e. subtracting the mean up-sampled feature over the dataset. This is demonstrated in Fig. C3, where we take the features on the broom point (green point on the first row) and measure its cosine similarity to all feature points of other images. We observe that LoftUP [S19] features provide very sharp and detailed maps, which also matches our motivation of using an up-sampler. However, we see that the point on the broom exhibits a baseline similarity with other non-relevant pixels (see bottom three non-relevant images). This is well addressed when performing a centering on the features (subtracting the global dataset mean). We also see that more recent up-samplers (AnyUp [S37] on the right) may not exhibit such baseline similarity. Note that our $C^2$-score would directly benefit from any advancements on foundation models [S31], as well as feature up-sampling methods [S37].

**Random Baseline.** A key part of $C^2$-score is the inclusion of random baseline, which allows to evaluate concepts while taking into account the baseline similarity of the probe dataset. For every image in the evaluating set, the random baseline samples a spatial attribution map with a random threshold (both from a uniform distribution). The consistency score Eq. (15) of random concept was at 0.0005 ($\sigma$=8e-6), averaged across three seeds, while on a per-class setup it was at 0.4008 ($\sigma$ =0.1166), averaged across the 1000 classes. This agrees with the original assumption that when the evaluation set is restricted to a single category, all the output features of the foundation model at use (here DINOv2) become more similar. Hence the $C^2$-score tackles this by considering the difference with respect to the random baseline.

# D  FaCT Provides Diverse Concepts

In the paper we demonstrated how our method provides shared concepts with high variety in spatial size Fig. 5. In Sec. D.1, we extend the Fig. 5 from the paper and additionally provide insights on the class-specificity of our concepts. Afterwards, Sec. D.2 demonstrates more qualitative samples across layers and architectures.

## D.1  Diversity in class-specificity and spatial size.

In Fig. D1 we report diversity of our concepts in terms of class-specificity (through concept-label entropy) and also in terms of spatial size. We also provide sample concepts to further demonstrate the diversity qualitatively.

**Concept-label Entropy [S25]** is defined for a concept $k$, activating on image $I$ with value $S^{k,I}$ as follows:

$$H_k = -\sum_{l=1}^{L} p_k(l) \log p_k(l) \,, \tag{D.1}$$

with $p_k(l)$ corresponding to aggregate activation mass of concept $k$ on samples with label $y_i = l$ over $L$ total unique labels.

$$p_k(l) = \frac{\sum_{i:y_i=l}^{N} S^{k,I}}{\sum_{i}^{N} S^{k,I}} \,. \tag{D.2}$$

Therefore, a class-specific concept that only activates for a single class would be assigned an entropy of 0, while a concept that uniformly activates across images of different labels, would be assigned an entropy of $log(1000) \approx 6.9$.

In Fig. D1 (upper left), we observe that both DenseNet and ViT models have concepts of different class-specificity, even at penultimate blocks (red). We see that for DenseNet, as we go to an earlier block, the portion of shared, higher-entropy, concepts increases (yellow). We further demonstrate samples (a-d) of different class-specificity. We observe that, e.g., concept (a) corresponds to 'Fox head', which is exclusive to two classes, while concept (b), 'Animal eyes' are shared across many species, resulting in higher entropy. In concept (c) 'Fur' and (d) 'White shirt', we similarly observe a high degree of sharing across classes. This is also evident within the top activating images coming from a variety of classes. The existence of such diverse set of concepts is in clear contrast to prior assumption on concepts being class-specific [S12, S24, S33].

**Spatial Size** is defined as the number pixels one needs to cover the top 80% of the input-resolution positive attribution of a concept (i.e. 80th percentile), averaged over the dataset. In Fig. D1 (upper right) we observe that our concepts come from a wide range spatial extents. This is in clear contrast to prior work [S12, S28] which assumes concepts to be fixed-sizes patches. Our visualized samples in Fig. D1 further validate this by having smaller concepts such as (a) 'Fox head' and (b)'Animal eyes', as well as larger spatial concepts such as (c) 'Fur'. The diversity in spatial size is further demonstrated in Figs. D2 and D3 for both architectures and across the layers.

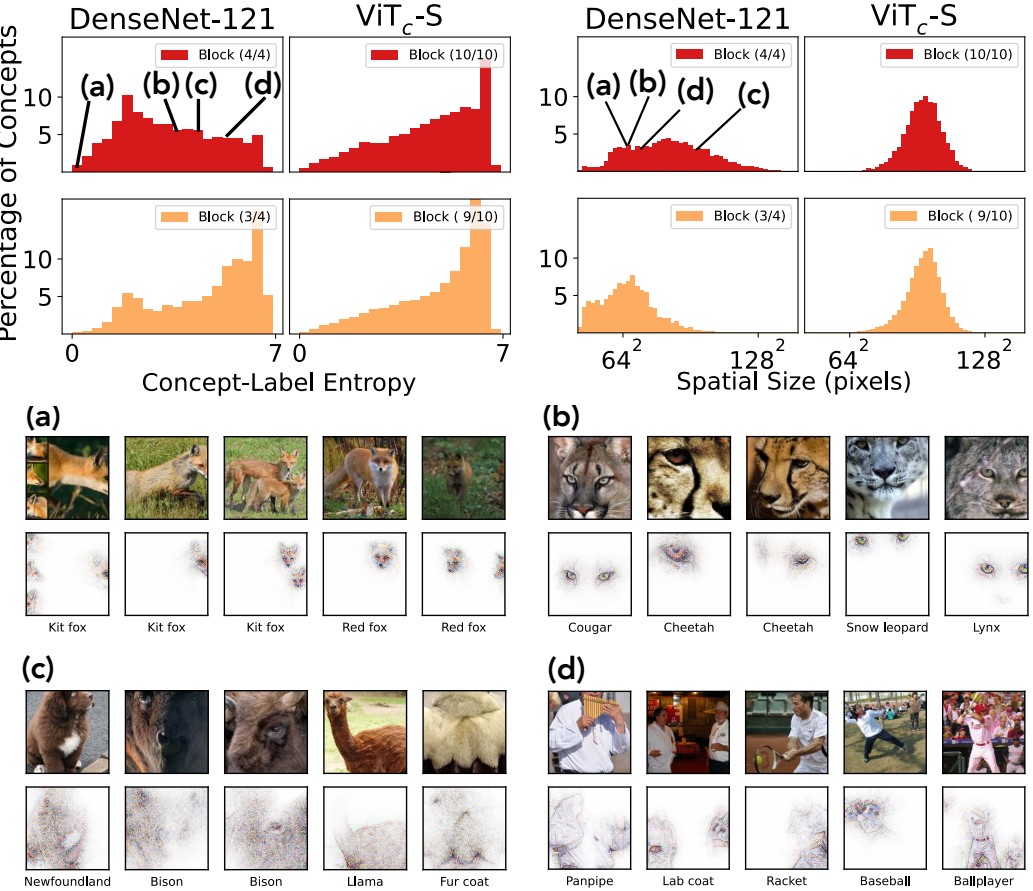

Figure D1: **FaCT offers diverse concepts.** We observe that our concepts, both for intermediate and late layers, come from a diverse distribution of class-specificty (upper left) and spatial size (upper right). We validate the plots by showing samples from different bins (a-d). We see class-specific concepts such as (a)'Fox head' and more class-agnostic concepts such as (c)'Fur' and (d)'White shirt'. For more qualitative results from both architectures, see also Figs. D2 and D3.

## D.2 Generalizing across layers and architectures

In Figs. D2 and D3 we show how FaCT generalizes across early and late layers, as well as across architectures (CNNs and ViTs). The displayed concepts additionally extend the observations made in Fig. D1, demonstrating the diversity of concepts in terms of spatial extent and class-specificity.

In Fig. D2, we generally find simpler concepts for earlier layers, such as 'Red dots', 'Green background field', 'Curves', 'Tiles', and 'Small animal legs', from top to bottom. We also observe a mix of high-level and simpler concepts for later layer (right col.), such as 'Baby faces', 'Shoreline', 'Red surface', 'Archs', and 'Ping pong ball'. These concepts also serve as counter examples for what prior work considers a concept to be. Unlike [S15, S12, S24] many of our concepts are shared across class, while class-specific ones such as 'Ping-pong ball' on the bottom right also exist. Our concepts are also not confined to be object- or part-centric [S4, S28], e.g. 'Shoreline' and 'Archs'.

In Fig. D3, we further demonstrate how FaCT generalizes across ViT architecture with interpretable concepts. We again see a mix of class-specific concepts, such as 'Junco beak' (first col. 4th row) and 'Saint Bernard brown fur' (second col. 5th row), together with shared concepts such as 'Bright yellow' (first col. 5th row) and 'Branches' (second col. first row). We observe that FaCT is able to encode small part-based (e.g. 'Sharp ears' first col. 3rd row) and large scene-based (e.g. 'Tiles' second col. 4th row) concepts, all being faithfully visualized at input. This is in contrast to prior work [S4, S28, S12] which assumes parts to be small patches and visualizes concepts by upsampling low-resolution intermediate maps, that does not generalize for ViT architectures.

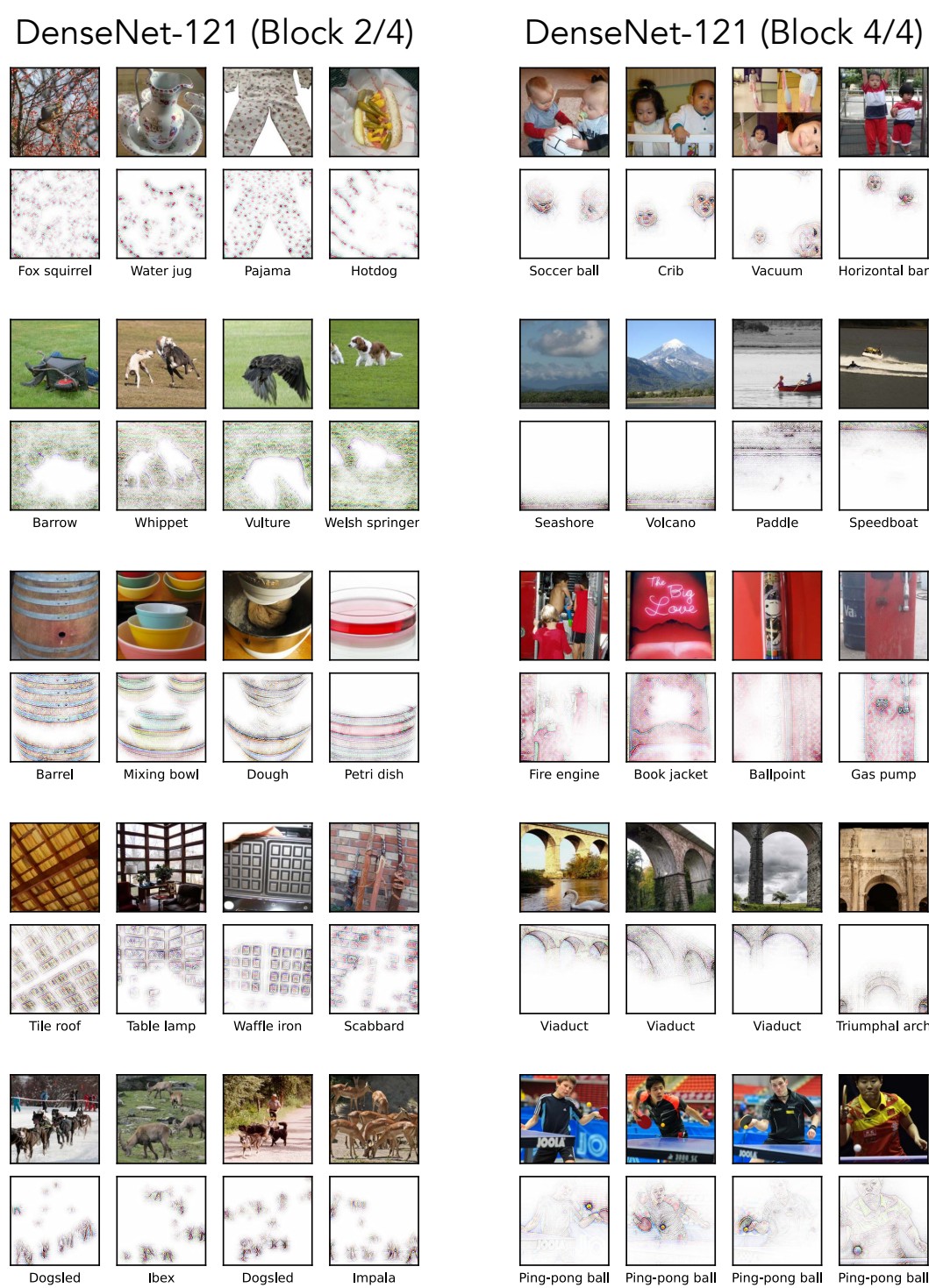

Figure D2: Sample concepts from early (left column) and late (right column) concepts of our DenseNet-121 FaCT models. We observe simple concepts such as 'Red dots', 'Green Background Field', 'Curves', 'Tiles', and 'Small animal legs' for early layer and concepts of higher semantics such as 'Baby Face','Shoreline','Red Surface', 'Archs', 'Ping-pong ball' for late-layer concepts. See also Fig. D3 for ViT concepts.

## ViT$_c$ - S (Block 9/10)   ViT$_c$ - S (Block 10/10)

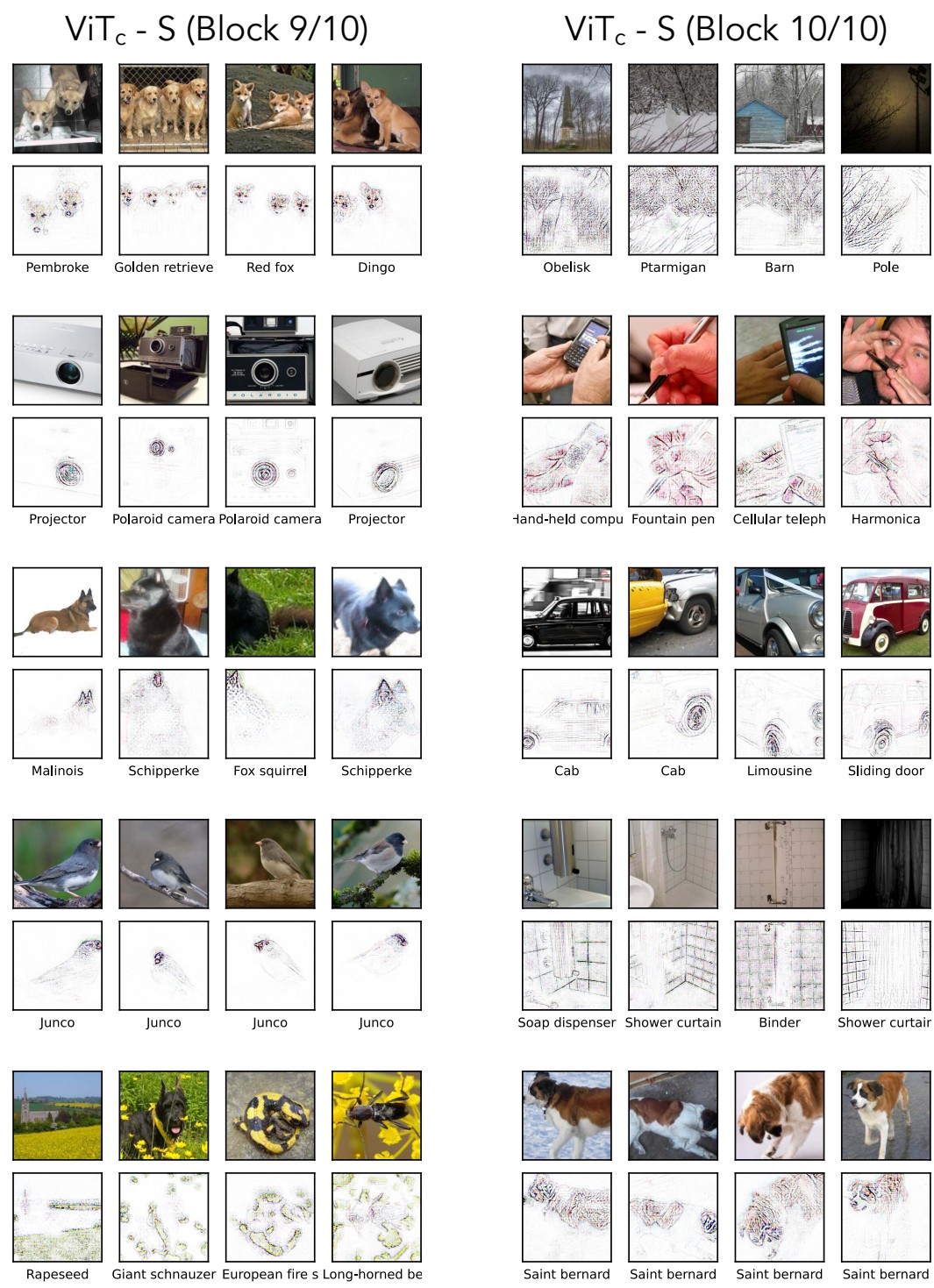

Figure D3: Sample concepts from Block 9 (left column) and Block 10 (right column) of our ViT FaCT model. Similar to Fig. D2 for DenseNet, here we also observe a mix of concepts of different spatial size and class-specificity. On the left column, we see 'Dog faces', 'Lens', 'Sharp dark ears', 'Junco beak', and 'Bright yellow', and on the right column we ee 'Branches', 'Human hands', 'Car wheels', 'Bathroom Tiles', and 'Saint Bernard's fur'.

# E   Details on the User Study

In this section we provide further details on our user study. As discussed in Section 5.3 in the main paper, we randomly sampled 100 late and 100 early concepts of our proposed FaCT model for DenseNet-121. We also randomly sampled 30 late and 30 early B-cos channel concepts as baseline. For each concept, we visualized top-10 images with highest activation together with their input attribution, as derived in Eq. (12). We additionally performed a counterbalanced AB/BA test, were for each of our 200 (early and late) concepts, we additionally plotted the top-10 images without any input attribution. The resulting 460 questions were then distributed to 10 randomized groups, such that each group has samples from both early- and late- FaCT and B-cos concepts. For the controlled study we made sure that no user sees both experimental (with input-attribution) and control (without) version of the same concept.

Our anonymous survey received in total 38 complete responses. At the beginning we instructed the participants with a set of sample concepts that one would consider interpretable and uninterpretable, accompanied with explanation on why, see Fig. E1.

After reading the introduction, the participants would then be randomly assigned to one of the 10 groups with 46 questions. See Fig. E2 for screenshots of two sample questions from two groups.

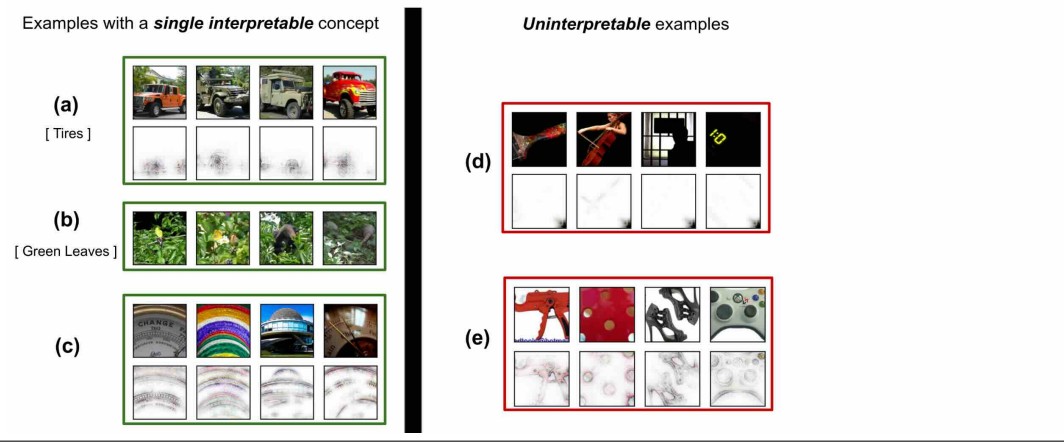

Figure E1: Screenshot of initial introduction provided to the user. We instructed the users with a general definition of a concept, followed by sample interpretable and uninterpretable examples with explanation. For sample questions, see Fig. E2.

.

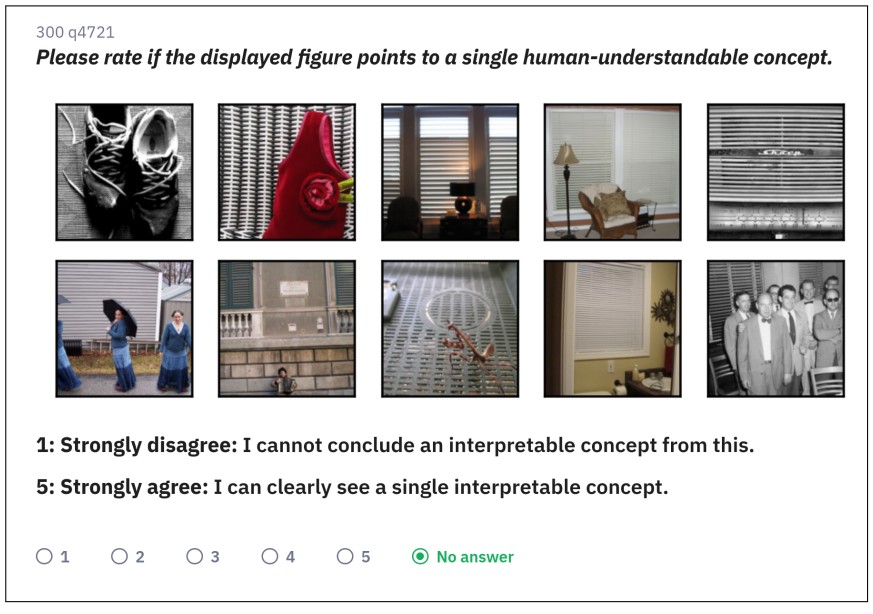

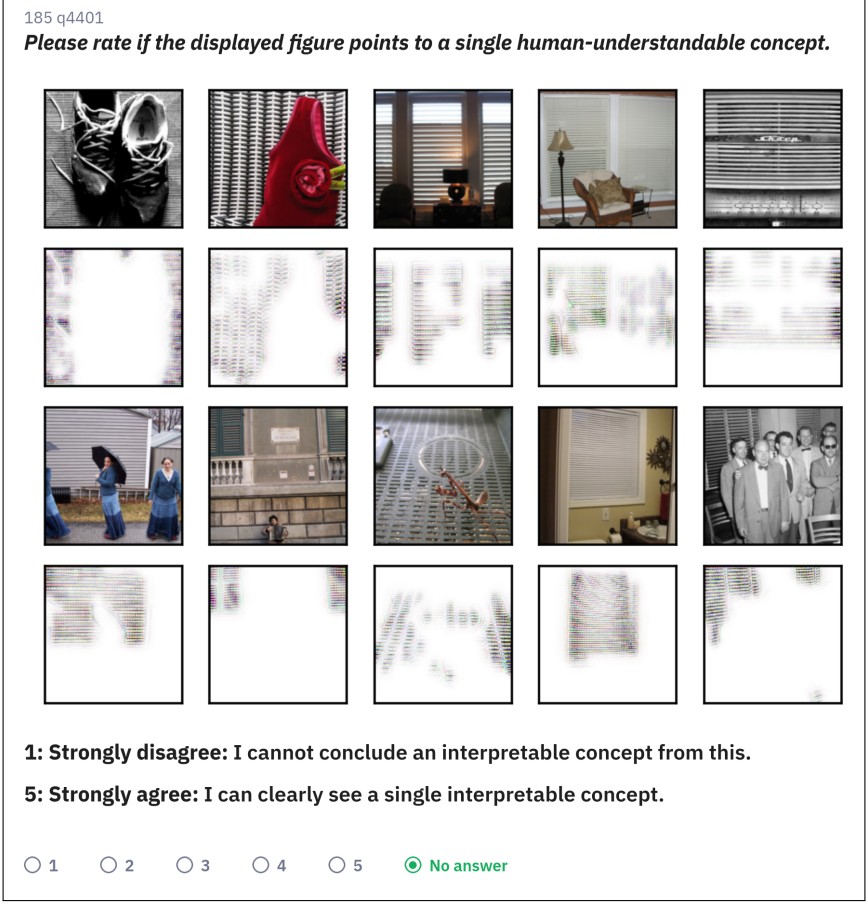

Figure E2: Screenshot of two sample questions from randomized **group 8 (top)** and **group 4 (bottom)**. The sample at the top (without input attributions) serves as a control sample for the one at the bottom.

# F  Details on Concept Deletion

We evaluated our concept contributions (Eq. (9) in the main paper) using concept-deletion metric [S11]. Given a concept importance measure, this evaluation framework deletes concepts in the order of most to least important and measures the changes of the target logit. If the importance measure is accurate, one would expect a sharper drop in the logit as the concepts are removed. In our case, since our concept-basis is shared across classes and are used for the final prediction, we additionally plot the drop in overall accuracy. In order to compare to prior work [S15, S12, S24], we evaluate their used concept importance measures, namely Saliency [S32] and Sobol indices [S10]. Note however that the comparison has to be done on the same concept set of the same layer, so that one only compares the difference in importance measures, without adding confounders such as different number of concepts or different distribution of significance over concept bases.

We evaluated the importance measures over 50 randomly selected classes of ImageNet, with 8 samples per class chosen from the test set. We based our implementation of Saliency and Sobol indices on definitions in [S11]. For Sobol indices [S10], we used Janson Estimator [S22], similar to CRAFT [S12]. While CRAFT configures Janson Estimator to 32 designs for the few per-class concepts, we were only able to use 4 designs, given the high number of concepts $K \in [8192, 16384]$, which still created more than 32,000 concept perturbation masks and forward passes *per image*. This further highlights how some of the importance measures tailored for setups with few class-specific concepts may be difficult to scale to large and shared concept bases.

**Additional Results for ResNet-50.** In Fig. F1 we provide additional results for ResNet-50 (similar to Fig. 7 in the main paper for DenseNet-121). We observe consistent trend as in Fig. 7, with our concept contributions (Eq. (9)) outperforming existing importance measures, for both logit and accuracy drop. We particularly see larger difference for earlier layers, with a larger gap between the curves.

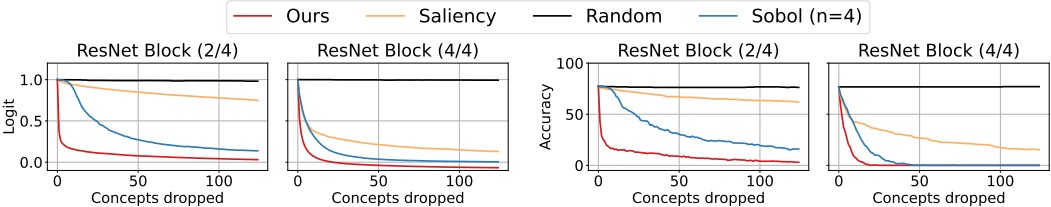

Figure F1: Additional Concept Deletion Results for ResNet-50. We observe that our proposed concept contributions (Eq. (9)) outperform existing concept importance measures with a sharper drop both in terms of top-1 logit and overall accuracy. This is similar to Fig. 7, but for ResNet-50.

**'Always-on' Concepts.** As discussed in Section 6 in the main paper, we observed a very sharp drop for B-cos contributions in Fig. 7 of the main paper (and in Fig. F1 for ResNet). In both cases, we found these concepts to be a small set that occur on 100% of the samples. This also matches the highly-frequent concepts observed in training the SAEs (see Sec. A). We hypothesize that these are mean feature vectors that are used for reconstruction in every sample. The sharp drop in Figs. 7 and F1 is caused when these concepts get deleted. Nevertheless, as we use the same concept-set for evaluating the concept-importance methods, this still shows that B-cos contributions (Eq. (9)) are best in identifying the most impactful concepts. To investigate the identification of most relevant concepts beyond these "always-on" concepts, in Fig. F2 we re-evaluated the concept deletion at early layers without removal of these few latents, only evaluating on the rest of the concept set. We see that the sharp drop disappears, yet Eq. (9) still outperforms other concept-importance measures.

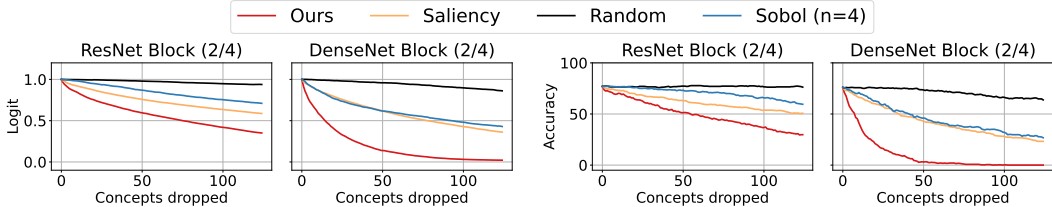

Figure F2: Similar to Figs. 7 and F1, we performed concept-deletion at early layers of ResNet and DenseNet, but without allowing the few 'always-on' concepts to be deleted. We observe that our proposed concept contributions (Eq. (9)) nevertheless outperform existing concept importance measures, with a sharper drop both in terms of top-1 logit and overall accuracy.

# G   Details on Inspecting Misclassification

In this section, we begin by providing further details on the Fig. 8 in the main paper and afterwards additionally compare the same confusion case for CRAFT [S12], a prior work with class-specific concepts, to show how our shared basis in Fig. 8 can provide additional insights.

In Fig. 8 of the main paper, we demonstrated the confusion between Basketball and Volleyball image with a set of concepts that mutually and exclusively contribute to both. To create this figure, we used DenseNet-121 FaCT model with concept-decomposition at Block 4/4. We then explained the Volleyball and Basketball logits individually in terms of concept contributions, using Eq. (9), leading to two contribution values per concept, for basketball and volleyball logits. For each concept-logit pair, we computed the contribution as the percentage of overall positive contribution of concepts. Finally, the 6 concepts shown in Eq. (9) were selected from the top-12 concepts that appeared for each logit.

**Comparison to class-specific methods.** Through Fig. 8 in the main paper, we were able to understand the confusion between the two logits Volleyball and Basketball on a concept-level, with concepts such as 'Ball' (A), 'Jerseys' (B), 'Man in sports-shirt'(C) and 'Limbs' (D) contributing to both logits. Below, we try to understand the same confusion case for ResNet-50 with a class-specific method CRAFT [S12]. In Fig. G1, we report the two Basketball and Volleyball logit for the same image in the middle, and show class-specific concepts that CRAFT offers for each on the side. While the concepts of CRAFT [S12] are indeed interpretable, we see that similar concepts repeat for each of the classes without any connection between the two, as opposed to our FaCT model that uses a shared basis with concepts contributing to both, e.g. the third and fourth row seem to both point to a similar 'floor' concept, yet this is not captured by the explanation method.

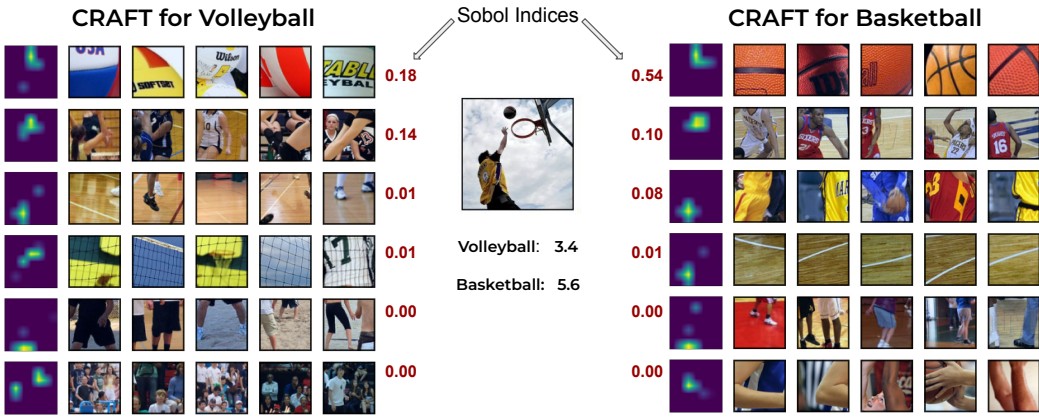

Figure G1: **Comparing class-confusion analysis with class-specific methods**. Here we plot CRAFT [S10] explanations for Volleyball (left) and Basketball (right) categories. Each row shows a concept with the up-samnpled concept attribution (left sub-cols) and Sobol indices (numbers in the middle) for each concept. While the concepts are quite interpretable, we see that they are repeated between the two classes (see the 'floor' concept on each side). This is in contrast to our setup, where a shared concept basis is used.

# H  Further Derivations

In this section we extend the derivations provided in Section 3.1 in the main paper. We begin by discussing why ReLU is a dynamic-linear transform, which was used in Eq. (6). We then demonstrate in Sec. H.2 how to derive input attribution of concept *contribution* as opposed to concept *activation* derived in Eq. (12) in the paper. Lastly, in Sec. H.3 we derive how the cross-layer contribtion of concepts to each other can be faithfully measured.

## H.1  Dynamic Linearity of ReLU

In the main paper, we discussed how in Eq. (6) our concept activations $\mathbf{U} \in \mathbb{R}^{H \times W \times K}$ is a dynamic-linear transformation of $F \in \mathbb{R}^{H \times W \times C}$. While our SAE definition in Eq. (7) is bias-free, it still uses a ReLU non-linearity, which in this section we explain how it can be considered as a dynamic-linear transform. For any tensor $\mathbf{X} \in \mathbb{R}^{D_1 \times D_2 \times D_3}$ a ReLU operation can be formulated as an element-wise multiplication with a tensor $\tilde{\mathbf{W}}(\mathbf{X}) \in \mathbb{R}^{D_1 \times D_2 \times D_3}$ of the same shape, such that

$$\text{ReLU}(\mathbf{X}) = \tilde{\mathbf{W}}(\mathbf{X}) \cdot \mathbf{X} \quad \text{s.t.} \quad \tilde{\mathbf{W}}(\mathbf{X}) = \begin{cases} 1 & \text{if } \mathbf{X} \geq 0 \\ 0 & \text{if } \mathbf{X} < 0 \end{cases}. \tag{H.1}$$

In fact, any piece-wise linear function can similarly be considered as dynamic-linear, but not vice versa (e.g. B-cos transforms [S2] in Eq. (3) in the main paper are dynamic-linear but not piece-wise linear). Therefore Eq. (H.1) shows that our concept activations defined in Eq. (6) are a dynamic-linear transformation of features, which allows our derivations for concept contributions in Eq. (9) and concept-attributions in Eq. (12) to hold true.

## H.2  Faithfully attributing concept *contribution* at input

In the main paper, we demonstrated how the *activation* of a concept for an image can be faithfully attributed to the input ( Eq. (12) in the paper). Note that in Eq. (10), the activation of a concept is defined by a summation over spatial dimensions. An output logit, however, may not rely on every spatial position equally. This is evidenced by Eq. (9), where the logit is a dynamic-linear combination of concept activations (both over spatial dimensions $H, W$ and concepts $K$). For convenience, we repeat the Eq. (9) below.

$$L_{\hat{c}}^{\text{FaCTs}} = \cdots = \sum_{k}^{K} \sum_{i,j}^{H,W} \tilde{\mathbf{W}}(\mathbf{U})_{i,j,k} \mathbf{U}_{i,j,k} = \sum_{k}^{K} \text{Contribution}_{k}^{\hat{c}}. \tag{H.2}$$

Therefore, for a particular concept $k$ contributing to logit $\hat{c}$, the $\text{Contribution}_{k}^{\hat{c}}$ can be formulated as follows:

$$\text{Contribution}_{k}^{\hat{c}} = \sum_{i,j}^{H,W} \left[\tilde{\mathbf{W}}(\mathbf{U})\right]_{k} \mathbf{U}_{i,j,k}. \tag{H.3}$$

Analogous to Eq. (12) in the main paper, we can now further decompose the $\mathbf{U}$ tensor as a dynamic-linear transform of input pixels to obtain a faithful input attribution of concept *contribution*. The only difference here is that we have a weighted sum of spatial positions (i.e. $[\tilde{\mathbf{W}}(\mathbf{U})]_{k}$) instead of uniform summation at the beginning of Eq. (12) in the main paper.

$$\text{Contribution}_{k}^{\hat{c}} = \sum_{i,j}^{H,W} \left[\tilde{\mathbf{W}}(\mathbf{U})\right]_{k} \mathbf{U}_{i,j,k} = \sum_{i,j}^{H,W} \left[\tilde{\mathbf{W}}(\mathbf{U})\right]_{k} \left[\text{ReLU}\big(\text{conv}(\mathbf{W}, F)\big)\right]_{i,j,k} \tag{H.4}$$

$$= \sum_{i,j,c}^{H,W,C} \left[\left[\tilde{\mathbf{W}}(\mathbf{U})\right]_{k} \tilde{\mathbf{W}}(F) F\right]_{i,j,c} = \sum_{i,j,c}^{H,W,C} \left[\tilde{\mathbf{W}}(F) f_{1 \to l}(I)\right]_{i,j,c} \tag{H.5}$$

$$= \sum_{i,j,c}^{H,W,C} \left[\tilde{\mathbf{W}}(F)\big(\tilde{\mathbf{W}}_{1 \to l}(I) I\big)\right]_{i,j,c} = \sum_{i,j,c}^{H_0,W_0,3} \left[\tilde{\mathbf{W}}(I) I\right]_{i,j,c}. \tag{H.6}$$

We therefore see that similar to Eq. (12) where the *activation* of a concept could be faithfully attributed to the input, the *contribution* of a concept to a particular logit can also be faithfully attributed to input.

In our experiments, we did not find distinguishable difference in visualization of concept activation and concept contribution. Throughout the paper we therefore consistently visualized the activation of concepts, i.e. Eq. (12) in the main paper.

### H.3 Faithfully measuring cross-layer concept-contribution

In Eq. (9) in the main paper, we demonstrated how every individual logit can be faithfully decomposed as a summation of concept contributions. If one considers a logit as a 'neuron', then essentially, Eq. (9) explains how a late-layer neuron (concept activation or logit) can be explained as contribution of earlier concepts. Thus one can use Eq. (9) for cross-layer contribution, by simply replacing the initial logit with a concept activation. Below we will nevertheless further derive this in detail.

Suppose we have a model $f(x) = f_{g \to n} \circ f_{l \to g} \circ f_{1 \to l}(x)$, where we have two early and late concept decompositions of the outputs at layers $l$ and $g$, named $\mathbf{U}^l \in \mathbb{R}^{H \times W \times K}$ and $\mathbf{U}^g \in \mathbb{R}^{H' \times W' \times K'}$, respectively.

$$F^g \approx \breve{F}^g \quad \text{s.t.} \quad \breve{F}^g = \text{conv}(\mathbf{V}^g, \mathbf{U}^g) = \text{conv}(\mathbf{V}^g, \text{ReLU}(\text{conv}(\mathbf{W}^g, F^g))) \tag{H.7}$$

$$L_{\text{FaCTs}} = f_{g \to n}(\breve{F}^g) = f_{g \to n} \circ \text{conv}(\mathbf{V}^g, \text{ReLU}(\text{conv}(\mathbf{W}^g, F^g))), \tag{H.8}$$

with intermediate feature tensor $F^g$ itself being a function of earlier features $F^l$:

$$F^l \approx \breve{F}^l \quad \text{s.t.} \quad \breve{F}^l = \text{conv}(\mathbf{V}^l, \mathbf{U}^l) = \text{conv}(\mathbf{V}^l, \text{ReLU}(\text{conv}(\mathbf{W}^l, F^l))) \tag{H.9}$$

$$F^g = f_{l \to g}(\breve{F}^l) = f_{l \to g} \circ \text{conv}(\mathbf{V}^l, \text{ReLU}(\text{conv}(\mathbf{W}^l, F^l))). \tag{H.10}$$

Therefore, the activation of concept $c$ from the late concepts $\mathbf{U}^g$ can be explained as a summation of contributions from early-layer concept activations $\mathbf{U}^l$:

$$\text{Activation}_c^g = \sum_{i,j}^{H',W'} \mathbf{U}_{i,j,c}^g = \sum_{i,j}^{H',W'} \text{ReLU}(\text{conv}(\mathbf{W}^g, F^g))_{i,j,c} \tag{H.11}$$

$$= \sum_{i,j}^{H',W'} \text{ReLU}(\text{conv}(\mathbf{W}^g, f_{l \to g}(\breve{F}^l)))_{i,j,c} = \tag{H.12}$$

$$= \sum_{i,j}^{H',W'} \text{ReLU}(\text{conv}(\mathbf{W}^g, f_{l \to g} \circ \text{conv}(\mathbf{V}^l, \mathbf{U}^l)))_{i,j,c} \tag{H.13}$$

$$= \sum_{i,j,c'}^{H,W,C} [\tilde{\mathbf{W}}(\mathbf{U}^l)\mathbf{U}^l]_{i,j,c'} = \sum_{\hat{c}}^{C} \text{Contribution}_{\hat{c}}^c. \tag{H.14}$$

where $\text{Contribution}_{\hat{c}}^c$ denotes the contribution of early-layer concept $\hat{c}$ at layer $l$, to the activation of late-layer concept $c$ at layer $g$. This is the exact derivation that was used for Fig. 1 in the main paper, with the early-layer 'Curve' concept contributing 2.9% to the activation of late-layer 'Wheel' concept.

# I  Computation Overhead

As discussed in Section 3.1, FaCT leverages Sparse Auto-Encoder as a bottleneck during the forward process, ensuring that the model only relies on the concept activations. To ensure that this does not result in significant computational overhead, in Table I1 we measured the time it takes for FaCT to process the entire ImageNet's validation set (i.e. 50,000 images) for DenseNet-121 at different layers. We compare the inference time with the corresponding B-cos model of the same depth. The results were averaged across three runs. We find FaCT to be quite comparable to the original B-cos architecture. Specifically, we observe less than 0.2 milliseconds inference-overhead per-image, which is likely to reduce further with inference optimization (e.g. changing the SAE-hooks to fixed layers and using 'torch.compile').

| Method | B-cos | FaCT @ Block 2/4 | FaCT @ Block 3/4 | FaCT @ Block 4/4 |
|---|---|---|---|---|
| Time (seconds) | 104 | 112 | 108 | 112 |

Table I1: Time required to process the entire ImageNet's test set for DenseNet-121 models (averaged over three runs). FaCT has very comparable inference-time compared to the B-cos model.

# J  Stability of SAE Training

As discussed in Section 3.1, FaCT trains Sparse Auto-Encoder to form a concept-basis. While different SAE training sessions may result in different final models, in general for the same layer we observed many concepts being repeated across configurations as well as across architectures at similar depth. To verify this quantitatively, we evaluated the recently proposed Stability Score [S13] for the same layer and dictionary size, but with different (Top-K) sparsity factor. We did this both for early and late-layer decompositions of DenseNet-121 and computed the pair-wise stability-score [S13] for (Top-K $\in \{8, 16, 32\}$ experiments. We observed score of 0.76 and 0.70 for Block 2/4 and Block 4/4 of DenseNet-121 models. While being trained on different models, datasets, and layers, our stability scores are quite on par with the ones in Table 1 of [S13], where the authors report 0.5 for the TopK-SAE under different seeds.

Additionally, we would like to point out that FaCT would directly benefit from new variants of SAEs, e.g. Archetypal SAEs in [S13], that may be introduced by the community. We would also highlight that with faithful input-attribution of concepts, as discussed in Eq. (12) of the main paper, FaCT allows the community to evaluate different concept-discovery methods (e.g. different SAE variants) under the lens of faithful attribution.

# K  Results for CUB Dataset

Throughout the paper, we put particular emphasis on having a scalable setup which remains competitive on ImageNet [S30] (see Section 5.1 in the main paper). In this section, we demonstrate how FaCT generalizes to other datasets, namely CUB-200 dataset [S30] for fine-grained classification. While there are many works that propose tailored models for this dataset [S4, S7, S35, S28], which have shown to often not scale to more challenging datasets [S33], our focus is to obsever whether FaCT remains competitive while providing interpretable *shared* concepts.

Following our ImageNet setup from the main paper, we thus trained FaCT on early, middle, and late blocks of a B-cos ResNet-34 (pre-trained on uncropped CUB). We collected the uncropped training set's features similar to Sec. A and trained our bias-free TOP-K SAEs with learning rate of 0.001, total concepts $K \in [1024, 2048]$, and sparsity factor TOP-K $= 16$.

We observe that across the layers, FaCT is able to maintain high accuracy while providing consistency gains. In particular, with less than 1% accuracy drop, we observe significant consistency gains for Block 3/4 ($0.24 \rightarrow 0.37$) and Block 4/4 ($0.32 \rightarrow 0.58$). For reference, standard ResNet-34 [S34], ProtoPNet [S4], and Deformable ProtoPNet [S7] are all below 77% accuracy, according to [S7], though the pre-training recipes may not exactly match. Nevertheless, we observe FaCT to remain competitive and provide more consistent concepts.

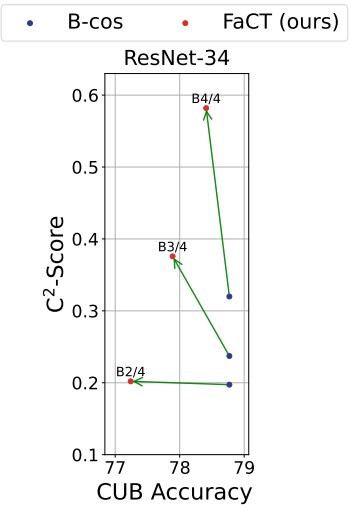

Figure K1: CUB results for ResNet-34 across layers.

When inspecting the concepts, we found the results to agree with what we observed for ImageNet in Section 5.1, with diverse set of shared concepts across the layers. In Figs. K2 and K3 we show concepts for Block 3/4 and Block 2/4. We observe many part-based concepts, in particular in Fig. K2 for Block 3/4, which are shared across classes. For earlier Block 2/4, i.e. Fig. K3, we observe lower-level concepts such as 'curves' or 'yellow-fur', while some also correspond to exact parts, e.g. 'wing edge' on the top-left. Interestingly, we saw an increased number of concepts for the background, which can be seen on the bottom rows of Fig. K3.

Our results thus show that FaCT, without having any assumption on the concepts being object parts, class-specific, or small patches, generalizes to other datasets. Having no restriction on concepts becomes more crucial when one moves to larger-scale datasets such as ImageNet, where the concepts required for the task may not necessarily correspond to parts, e.g. see scene-centric 'Shoreline' concept in Fig. D2 (second-row) or 'Bathroom Tiles' in Fig. D3 (fourth-row).

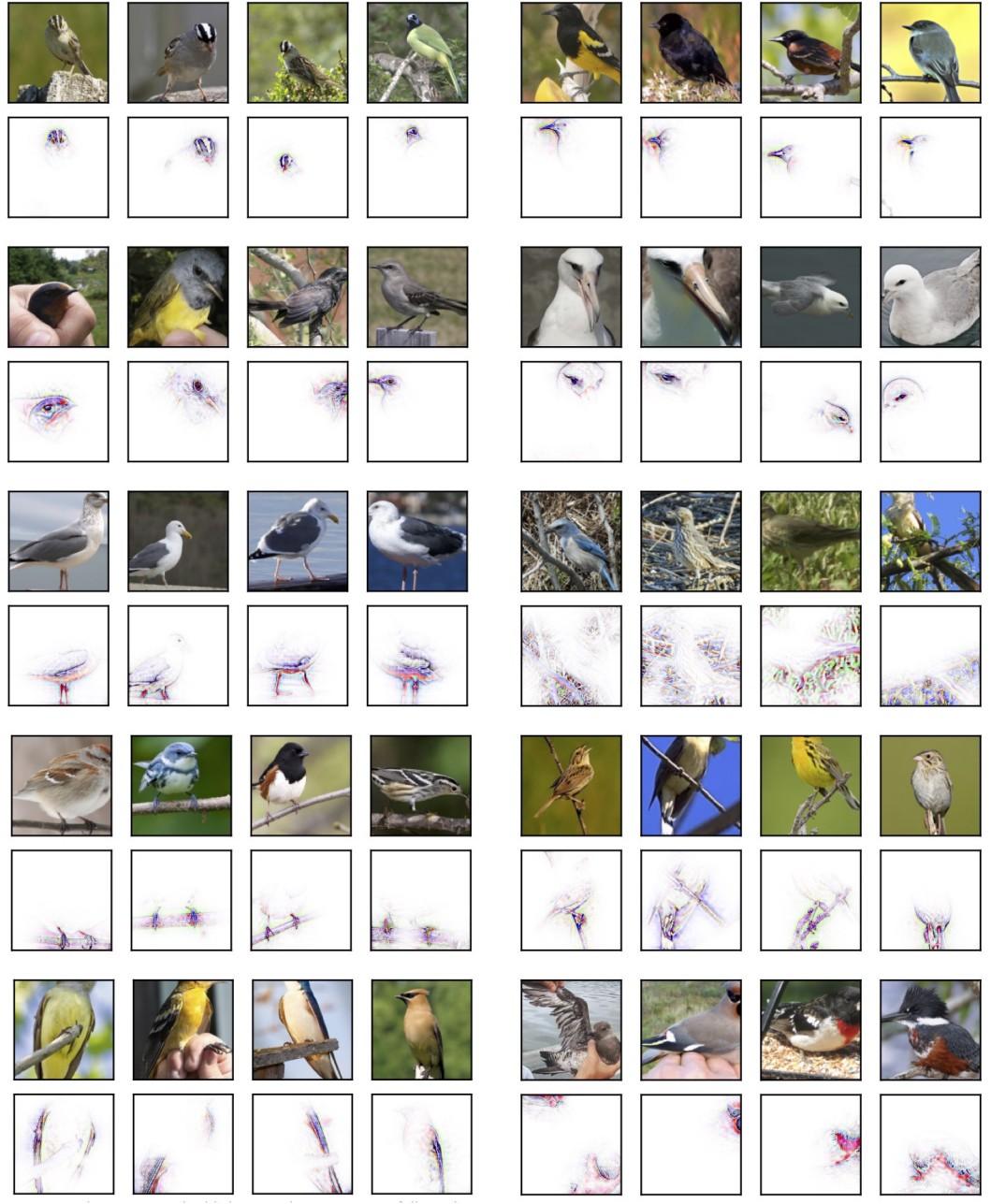

Figure K2: Sample FaCT Concepts from Block 3/4 on ResNet-34. We observe many corresponding to object parts, such as heads and beaks (top-two rows), legs and tails (rows three and four). Notice that many of these parts are shared among classes, e.g. 'legs on the branch' (fourth-row left).

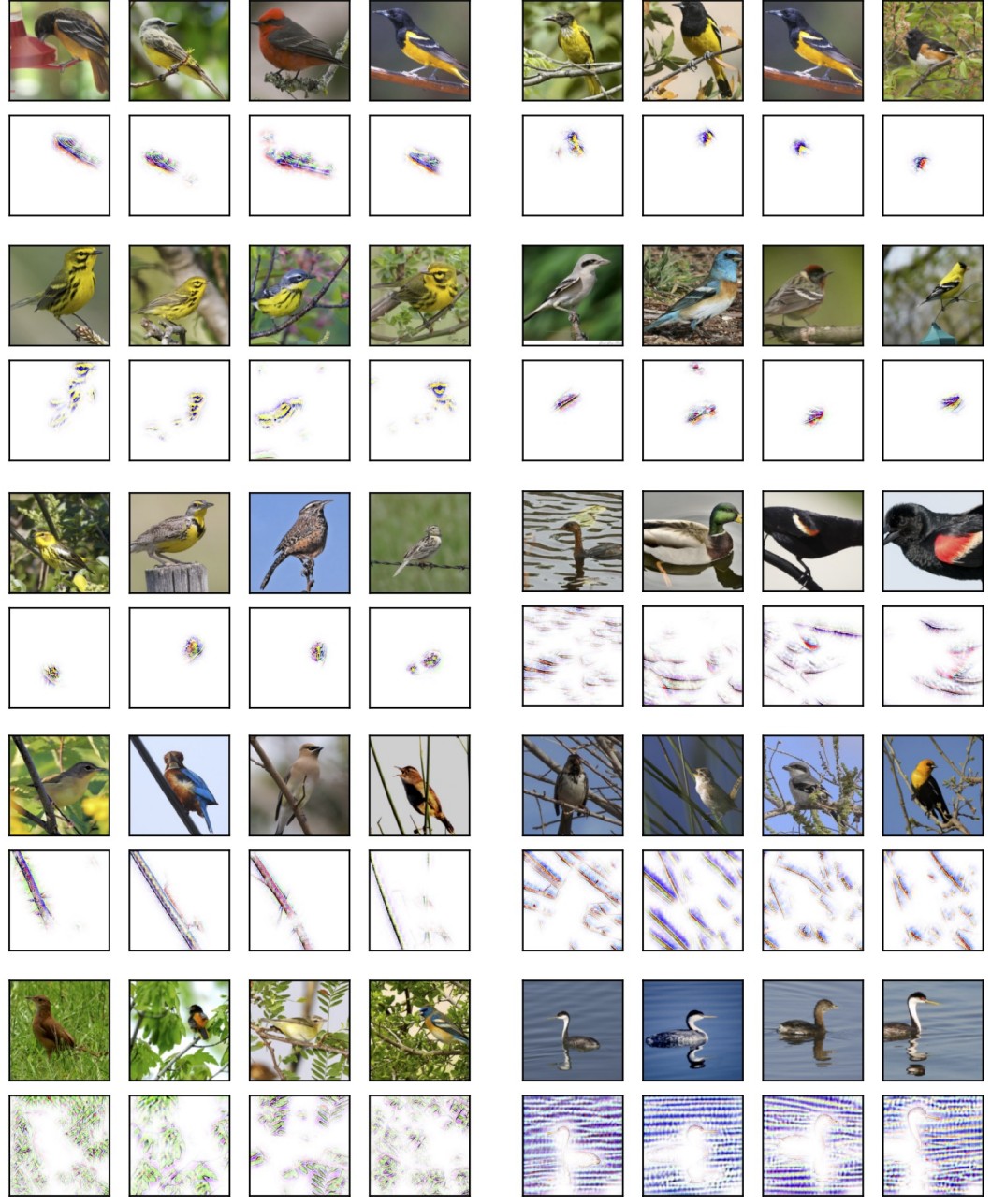

Figure K3: Sample FaCT Concepts from Block 2/4 on ResNet-34. We observed many concepts corresponding to simpler features, such as 'yellow-fur' (second-row left) or 'curves' (third-row right). We also observed an increased number of background concepts such as branches (fourth row) or water/land backgrounds (bottom row).

# L  Limitations and Future Work

In this work, we discussed a new model FaCT with inherent concept-based explanations with a concept basis shared across classes. Our proposed model can further explain every logit in terms of concept contributions Eq. (9), while faithfully attributing every concept to input Eq. (12). Such a faithful concept-based explanation however does not guarantee the *interpretability* of our concepts. Indeed, in Fig. 6 in the main paper, we demonstrated that our concepts are more interpretable than baselines, yet, we also see that there exist uninterpretable concepts with low scores from the users. We believe the next step would be further inspections on less interpretable concepts, their contributions to different predictions (through Eq. (9)). While uninterpretable concepts are less desirable for the end-user, whether a model without such concepts can be as performant as FaCT, is an open question. Perhaps a relevant direction could be a single-stage training paradigm, where FaCT is trained from scratch and is regularized towards more-interpretable concepts, as opposed to the two-stage training.

Further, our work leverages SAEs for arriving at the concept basis in an unsupervised manner. While SAEs offer the advantage of having concept activations as a linear (and in our case bias-free) transform of features, which we used for faithful input attributions in Eq. (12), we acknowledge the on-going discussion on SAEs, and note that our approach would also benefit from further research in this direction. For example, recently [S13] proposed a new paradigm of training SAEs by constraining the dictionary vectors to lie on the manifold of training features. Of course, this would directly help with training our FaCT models as well. In fact, we argue that one could now experiment different directions of training SAEs on our proposed model FaCT, so that the resulting concepts can better be studied, through faithfully visualizing them at input-level and faithfully measuring their importance to the final logits.

Additionally, we proposed a new concept-consistency metric $C^2$-score which leverages DINOv2 features for a class-agnostic concept consistency evaluation. While in Sec. C we demonstrated how this is superior to existing annotations and how it aligns well with our user study, a further study on what are the limitations of DINOv2 for such tasks across concepts of different semantics, and whether there exist better alternatives is indeed still an open question, which could significantly influence how concepts are evaluated in the future.

Lastly, the main ingredients of FaCT are the use of B-cos layers [S2, S3] for faithful attributions and SAE for concept extraction. This allows FaCT to readily extend to other modalities such as language, as both SAEs and B-cos layers have been shown to extend to other modalities [S21, S36]. Such applications would also allow exploring whether the faithful concept-based explanations of FaCT allow for better concept-editing for steering models [S5].

# M  Societal Impact

Our work puts great emphasis on the *faithfulness* of concept-based explanations and proposes a new model FaCT which can faithfully report existence of concepts and their significance to the prediction. It is therefore a step towards models with explanations that can be trusted for safety-critical applications, such as health domain, where the explanations should not mislead the user. We also proposed a novel metric $C^2$-score for assessing quality of concepts, which can benefit future concept-based methods as an automated evaluation tool.

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
