# OpenReview forum: "FaCT: Faithful Concept Traces for Explaining Neural Network Decisions"
_NeurIPS.cc/2025/Conference — NeurIPS 2025 poster_

### Official Review · Reviewer_qa8G · 2025-06-29

**Clarity:** 3
**Significance:** 3
**Originality:** 2
**Rating:** 5
**Confidence:** 5

**Summary:**

This study aims to understand the behavior of deep learning models at a global concept level, and unlike previous post-hoc concept-based approaches, it proposes an inherently interpretable model architecture that enables faithful explanations of model decisions. In addition, it introduces a novel evaluation metric called C²-score, designed to fairly compare different concept-based methods in a class-agnostic and annotation-free manner.

**Questions:**

- The proposed method builds on a pretrained and frozen B-cos network, with sparse autoencoders trained separately. Would it be possible to train the full FaCT architecture from scratch instead of relying on a pretrained B-cos network? Specifically, to obtain meaningful concept representations, could the model first be trained without the SAE modules during the initial epochs, and then continue training with the SAEs integrated, potentially enabling a single-stage training process instead of a strictly two-stage setup? Have the authors explored joint or progressive training strategies to this end?
- While the authors report that FaCT maintains performance competitive with B-cos, it is worth noting that B-cos networks themselves incur a 2–3% drop in accuracy compared to original models. Given that FaCT introduces an additional drop of up to 3%, the overall performance gap from the original architecture may approach 6%. This is a non-trivial trade-off in accuracy. Could the authors comment on how they justify this gap in practical scenarios, particularly when applied to real-world tasks?
- Finally, the proposed C²-score quantifies concept consistency as the difference between the consistency of a given concept and the consistency of a random concept. However, since the consistency of a random concept may vary across runs, would it not be more robust to compute the C²-score over multiple random seeds and report the mean along with its standard deviation? Have the authors examined the sensitivity of the C²-score to the randomness?

**Ethical Concerns:**

["NO or VERY MINOR ethics concerns only"]

**Final Justification:**

The authors have addressed most of the concerns. I think the paper has merits for the community.

**Limitations:**

- The semantic interpretation of each concept ultimately relies on human judgment, as the model does not assign explicit labels to the learned concepts.
- The method is only evaluated on a single dataset (ImageNet), limiting the assessment of its generalizability.

**Paper Formatting Concerns:**

No. There are no outstanding concerns.

**Quality:**

3

**Strengths And Weaknesses:**

Strengths
- This paper proposes a concept-based explanation method that is inherently interpretable model. Since concepts are explicitly used in the forward pass, model predictions are directly computed based on concept representations.
- The proposed approach is generalizable to both CNN and ViT architectures.
- The method achieves competitive performance on the ImageNet classification task while maintaining strong explainability compared to baseline models.
- The paper introduces a novel evaluation metric, C²-score, which can evaluate various concept-based methods without requiring additional annotations.

Weaknesses
- Although the model provides faithful visualizations of concepts, their semantic interpretation still requires human inspection, as the model does not assign explicit concept labels.
- The proposed method builds on a pretrained B-cos network with frozen weights, and introduces an additional training stage by separately learning sparse autoencoders, thereby breaking the end-to-end training pipeline.
- All experiments are conducted solely on ImageNet. Evaluations on other datasets would strengthen the claim of generalizability.
- As an inherently interpretable approach, the method should be compared with other inherently interpretable baselines such as Concept Bottleneck Models or Part-Prototype Networks.

---

> ### Author Rebuttal · Authors · 2025-07-31
>
> We thank the reviewer qa8G for their in-depth feedback and questions. We are very happy to see that the reviewer finds our approach generalizable, our model competitive and our metric novel. In the following, we will discuss the Weaknesses **(W1-4)**, Questions **(Q1-3)**, and Limitations **(L1-2)** raised by the reviewer.
>
> **Interpreting the Concepts (W1, L1):** We intentionally avoided using pre-defined concept sets that would bias the interpretation towards what we as humans expect the model should learn, and instead used an unsupervised concept-discovery method (SAE) for identifying the concepts the model finds useful. With FaCT, we aim at making the *concepts that the network learned* for decision-making transparent, which **do not need to align with pre-defined human concepts**. This allows us to maintain the performance of powerful models while allowing to interpret their reasoning. That said, we agree that having explicit names assigned to the concepts can benefit the interpretability for the end user, and have therefore followed this suggestion and used CLIP-Dissect [R1], which labels neurons (SAE latents in our case) using a CLIP model. Applying CLIP-Dissect on our concepts to generate the names would name the the concepts (A-F) shown in Fig. 6 to the following words (for each concept we show the top-3 matches):
>
>
> [A]: volleyball, rugby, balls
>
>
> [B]: jerseys, uniforms, players
>
>
> [C]: rugby, sergio, concentration
>
>
> [D]: gymnastics, acrobat, flexible
>
>
> [E]: volleyball, tournaments, serve
>
>
> [F]: basketball, layup, contested
>
> We find these names to be quite consistent with how we would interpret these concepts. We will add this discussion to the final revision.
>
> **Using a pre-trained network (W2, Q1):** We thank the reviewer for raising this point. There are two main reasons why we focused on training FaCT with pre-trained B-cos checkpoints.
>
> First, our current two-step procedure for training FaCT allows it to be readily applied to existing trained B-cos models, which is particularly important given the recent focus on extending B-cos networks to different domains such as Natural Language Processing [R2] and Self-supervised models [R5].
>
> Second, SAEs have been mainly proposed as an unsupervised concept-discovery method to be applied on top of a structured (trained) representation, with the sparsity constraints being the only prior guiding the model towards identifying concepts (i.e clusters). We thus suspect that training them on top of an unstructured representation may add to their instability or lead to less-interpretable latents.
>
> We do however agree that having the choice of one-stage training from scratch can be desirable in certain cases, and will add this in the future work discussion.
>
>
> **Other Datasets (W3, L2):** We thank the reviewer for their suggestion. We mainly experimented with ImageNet to ensure the competence of FaCT. Following your suggestion, we evaluated FaCT also on the (not-cropped) CUB dataset for ResNet-34 architecture. Similar to ImageNet experiments, we started with a pretrained B-cos checkpoint (ImageNet B-cos finetuned on CUB) and trained our SAEs at different layers, having the rest of the model frozen. The results can be seen in the table below.
>
>
> ### Table 5: Performance for ResNet34 on uncropped CUB dataset
> ### (Standard: 76.0 from [13])
> | Method | Accuracy | Consistency Gain (C2-Score) |
> | -  | -  | -  |
> | B-cos   | 78.8     | -   |
> | FaCT @Block 2/4     | 78.6     | 0.27 → 0.38   |
> | FaCT @Block 3/4     | 78.0     | 0.33 → 0.51   |
> | FaCT @Block 4/4     | 78.5     | 0.49 → 0.63   |
>
>
> We observe that FaCT works out of the box on smaller-scale datasets such as CUB and is able to maintain the accuracy (<0.8 % drop across). We thank the reviewer for their suggestion and will add the CUB experiments, together with qualitative results of the shared concepts for the final version.
>
> **Comparison to ProtoPNets and CBMs (W4):** As our main focus was large-scale ImageNet evaluations, we mainly compared with models that extend to ImageNet, such as B-cos channels and post-hoc methods such as CRAFT and CRP. Existing implementations of Part-Prototype often do not extend to ImageNet. In the two tables below, we compare a set of ProtoPNets and CBMs that had a comparable setup to ours.
>
> ### Table 6: ResNet34 comparison on CUB and ImageNet
> | Method | CUB Accuracy | ImageNet Accuracy |
> | - | - | - |
> | vanilla  | 76.0*  | 73.3   |
> | ProtoPNet   | 72.4* | 65.5**  |
> | Deformable ProtoPNet [13]  |  76.8*  | 63.4** |
> | B-cos | 78.8 | 72.3 |
> | FaCT @Block 2/4 | 78.6 | 71.3 |
> | FaCT @Block 3/4 | 78.0 | 70.4 |
> | FaCT @Block 4/4 | 78.5 | 71.0 |
>
>  *Numbers from [13], trained on uncropped CUB images ; **Numbers from [46]
>
> ### Table 7: ResNet50 comparison on ImageNet
>
> | Method             | ImageNet Accuracy |
> | -                  | -                 |
> | LF-CBM[34]             | 72.0***           |
> | Torchvision V1     | 76.1              |
> | Torchvision V2     | 80.2              |
> | B-cos              | 79.5              |
> | FaCT @Block 2/4    | 77.6              |
> | FaCT @Block 3/4    | 77.1              |
> | FaCT @Block 4/4    | 77.3              |
>
> ***Numbers from [34], using Torchvision V1 recipe
>
> Besides the significant performance advantage on CUB and ImageNet, FaCT offers a shared concept-basis, extends to different layers, and architectures (CNNs and ViTs), all while providing a faithful input-level attribution of concepts.
>
>
> **Performance Margin to Vanilla Models (Q2):** We would like to highlight that the recent V2 version of B-cos models [9] that we use obtain much closer performance to vanilla models. (e.g. only 0.8% drop on DenseNet-121). We would like to point to Table 5, 6, and 7 above. In all the three tables we observe that FaCT models are performing on par to standard baseline (Maximum 3.3% drop across all tables). We also observe that on a smaller-scale (CUB) dataset presented in Table 6, the performance gap compared to B-cos is even smaller (<0.8%). In the revision, we will add the vanilla baseline numbers to the comparisons for clarity.
>
> **Random Baseline in C2-Score (Q3):** We observed consistent results for the random baseline across runs. In particular, for multi-class evaluations, the consistency score of the random baseline was at 0.004 ($\sigma$=0.0005), averaged across three seeds. For the per-class setup, the random baseline score was 0.13 ($\sigma$=0.021), averaged across the random baselines of all 1000 classes. We will add these for the final revision.
>
> ---
> **[R1]:** CLIP-Dissect: Automatic Description of Neuron Representations in Deep Vision Networks, ICLR 2023
>
> **[R2]:** B-cos LM: Efficiently Transforming Pre-trained Language Models for Improved Explainability, preprint Feb 2025
>
> **[R5]:** How to Probe: Simple Yet Effective Techniques for Improving Post-hoc Explanations, ICLR 2024

---

> > ### Comment · Reviewer_qa8G · 2025-08-05
> > **Thanks**
> >
> > The authors have addressed all previous concerns comprehensively. The additions of concept labeling, new dataset evaluations, extended comparisons, and metric clarifications significantly strengthen the paper. The paper makes a solid contribution to XAI.

---

### Official Review · Reviewer_RaLA · 2025-07-02

**Clarity:** 4
**Significance:** 3
**Originality:** 3
**Rating:** 4
**Confidence:** 4

**Summary:**

This paper introduces a concept-based model, FaCT, that aims to ensure faithful concept attribution both to the output logits and the input image space. The model integrates B-cos networks and sparse autoencoders (SAEs) across intermediate layers to learn concept representations that are actively used in decision-making. Furthermore, the authors propose a new evaluation metric, C²-score, for concept consistency using DINOv2 features. The proposed FaCT model is validated through quantitative metrics, including ImageNet accuracy, concept deletion, C²-score, qualitative visualizations, and a user study.

**Questions:**

1. How sensitive is the concept decomposition to SAE hyperparameters?
2. Could FaCT be applied to other modalities (e.g., text)?
3. Can your concept contributions be used for counterfactual generation or concept editing?

**Ethical Concerns:**

["NO or VERY MINOR ethics concerns only"]

**Final Justification:**

Thank you to the authors for their thorough and detailed responses to my questions. After reviewing the authors' responses and considering the feedback from other reviewers, I am inclined to accept this paper.

**Limitations:**

Yes, the authors discussed the limitations in the Appendix. However, it would be better to move some of the discussions to the main text.

**Quality:**

3

**Strengths And Weaknesses:**

**Strengths:**
1. The proposed idea is technically sound and novel.
2. In contrast to post hoc methods, the proposed model inherently ensures the learned concepts are faithful to the decision-making process.
3. The proposed method is generalizable to different architectures, including CNN and ViT.
4. User study is provided to evaluate the actual quality of interpretations.
5. The paper is well-organized and easy to follow.

**Weakness:**
1. The learned concepts still lack human explicit human-grounded semantics. The labeling of the concepts rely on input feature attribution or image retrieval, which is subjective to the human annotator and not scalable.
2. The evaluations are primarily focused on ImageNet, which is mismatched with the concept-level interpretations offered by the proposed method. Evaluations on concept-level annotations, such CUB, Part-ImageNet, and PASCAL-part, will benefit this paper.
3. Missing a comprehensive ablation study on the hyperparameters. For example, the effect of SAE sparsity (top-k) and expansion factor (dictionary size) need to be studied systematically.

---

> ### Author Rebuttal · Authors · 2025-07-31
>
> We would like to begin by thanking reviewer RaLA for their detailed feedback and questions. We are happy to see the reviewer finds our idea novel, our method generalizable, and our writing easy to follow. In the following, we will address the Weaknesses **(W1-3)**, Questions **(Q1-3)** raised by the reviewer.
>
>
> **Human-alignment and Scalable Labeling (W1):** Please note that there is an inherent difference between explicit human-aligned models, such as CBMs, that are forced to learn a mapping to pre-defined concepts and conventional models, that learn concepts that they deem useful, which need not correspond to what we as humans would like to see or even comprehend. The advantage of the latter is obvious in terms of performance, achieving much higher accuracy than e.g. CBMs at the cost of being less transparent. With FaCT, we aim at making the **concepts that the network learned** for decision-making transparent, which **do not need to align with pre-defined human concepts**. This allows us to maintain the performance of powerful models while allowing to interpret their reasoning. Here, we believe a visualization of concepts will provide a better understanding of what is happening than text, given that we work on a vision task, which was also a key motivation for ProtoPNet, PipNet and others.
> Nonetheless, if textual interpretations are preferred, we can use off-the-shelf methods to label our concepts such as CLIP-Dissect [R1] which name individual neurons using a language-aligned model (i.e CLIP). To satisfy our curiosity, we applied CLIP-Dissect [R1] to our SAE latents, and got the following names for the concepts (A-F) shown in Fig. 6 of the paper: (for each concept we show the top-3 matches):
>
>
> [A]: volleyball, rugby, balls
>
>
> [B]: jerseys, uniforms, players
>
>
> [C]: rugby, sergio, concentration
>
>
> [D]: gymnastics, acrobat, flexible
>
>
> [E]: volleyball, tournaments, serve
>
>
> [F]: basketball, layup, contested
>
>
> These human-understandable textual names seem meaningful with respect to the concepts in Fig. 6. While not our primary focus, we will add this downstream application for textual explanations to our paper, thank you for raising this point.
>
>
> **Evaluation on other Datasets (W2):** We primarily avoided using fixed human annotations to evaluate our shared concept basis as the annotations are class-specific, subject to arbitrarily chosen granularity level, and are independent of what a model might learn to be useful for a decision. We would like to refer to Fig. C1 in  Appendix, where we show sample concepts that are not considered in Part-ImageNet annotations, while even the PCA visualization of DiNOv2 features seem to be better suited. We will make this comparison and discussion more prominent for the final version. We have also extended our evaluation to the CUB dataset, to ensure that our findings also hold on smaller-scale, but fine-grained classification (kindly refer to Table 3 in response to reviewer Lkv1).
>
> **Ablation on Hyper-parameters of SAE (W3):** We have described our hyper-parameter sweep and model selection criteria in (Appendix A, Lines 669-684). We have also plotted the models in terms of sparsity vs. accuracy in Fig. B1, where each point corresponds to a particular (top-k, expansion factor, and learning-rate) configuration. We had deferred these results to the Appendix in lack of space. We will make the references more prominent for the revision.
>
>
> **Sensitivity to Hyper-parameters of SAE (Q1):** In general for the same layer we observed many concepts being repeated across configurations as well as across architectures at similar depth. Following your question, we evaluated the recently proposed Stability Score [R4] for the same layer and dictionary size, but with different (top-k) sparsity factor. We did this both for early and late-layer decompositions of Densenet-121. While being trained on different models, datasets, and layers, our stability scores are quite on par with the ones in Table 1 of [R4], where the authors report 0.5 for the TopK-SAE under different seeds. We would also like to point out that FaCT would directly benefit from new variants of SAEs, e.g. [R4], that may be introduced by the community. We will extend the quantitative stability evaluation for the final revision.
> ### Pair-wise Stability Score
> ### Averaged over three pairs of experiments (TopK ∈ 8, 16, 32)
> | Method  | Stability Score |
> | -  | -  |
> | FaCT @Block 2/4  | 0.76  |
> | FaCT @Block 4/4  | 0.70  |
>
>
>
> **Applying FaCT to other Modalities (Q2):** The main components of FaCT are the use of B-cos layers (for faithful input-attribution and output-contributions) together with SAE for interpretable concept-based representation. Both of these have been recently explored for other domains such as natural-language processing. In particular, B-cos LM [R2], introduces a B-cos language encoder and evaluates the interpretability and faithfulness of attributions on the input sequences. Additionally SAEs [6, 11] were originally introduced for language models. We thus believe FaCT would lend itself well for other modalities and will add this to the future work discussion for the revision.
>
>
>
> **Concept-Editing (Q3):** Thank you for this interesting suggestion. Indeed one of the advantages of FaCT is that it uses concepts *as part of the decision-making process*, and hence modification to the concept representation can be explored and removing concepts does indeed affect the output behaviour of the model (e.g. Fig. 2). While B-cos models have shown to lend themselves well for being guided through their attributions, e.g. to avoid relying of spurious features (see Fig. 1 in [R3]), FaCT directly allows for steering the model *on a concept-level* (e.g. by explicitly masking certain SAE latents). We will add this as a future work direction in our revision.
>
>
> We will also move the limitations section to the main text in our revision.
>
> ---
> **[R1]:** CLIP-Dissect: Automatic Description of Neuron Representations in Deep Vision Networks, ICLR 2023
>
> **[R2]:** B-cos LM: Efficiently Transforming Pre-trained Language Models for Improved Explainability, preprint Feb 2025
>
> **[R3]:**  Studying How to Efficiently and Effectively Guide Models with Explanations, ICCV 2023
>
> **[R4]:** Archetypal SAE: Adaptive and Stable Dictionary Learning for Concept Extraction in Large Vision Models, ICML 2025

---

### Official Review · Reviewer_Lkv1 · 2025-07-03

**Clarity:** 3
**Significance:** 3
**Originality:** 3
**Rating:** 5
**Confidence:** 3

**Summary:**

The authors introduces FaCT, a model that makes neural network decisions easier to understand by breaking them down into concepts, like "yellow" or "wheel", that can be traced across all layers of the network. Different from prior art which don't truly reflect the model's internal reasoning, FaCT ensures that these explanations are true to how the model works and can be visualized clearly. The core idea is to use B-cos transformations which preserves linear contribution additivity and sparse autoencoders (SAEs) to present human-interpretable concept vectors. The paper also introduces a new metric called C2-Score (concept consistency score), which leverages DINOv2 features to quantify how consistent a concept is across different images. Experiments show the proposed method learned concepts more semantically consistent and human interpretable than the prior art.

**Questions:**

The paper would be stronger if the authors can address the following:
- While the paper shows how to visualize concepts, it doesn't resolve how to name or interpret each concept as some concepts are not easily aligned with human concept.

**Ethical Concerns:**

["NO or VERY MINOR ethics concerns only"]

**Limitations:**

yes

**Quality:**

3

**Strengths And Weaknesses:**

Strength
- The approach is novel in achieving complete, faithful decomposition of predictions into concepts across layers. Prior concept models
often provided only approximate importance of concepts to outputs, or needed additional tools to visualize concepts. FaCT’s design ensures by construction that all attributions are faithful and additive.
- FaCT’s concepts are shared among all classes and across layers, which results in a more compact and general explanation basis. A concept like “wheel” can contribute to recognizing buses, cars, etc.
- High accuracy: FaCT achieves competitive performance on ImageNet, which shows it doesn't compromise much predictive power to obtain interpretability.
- Comprehensive Evaluation, including a C2-score quantitative metric, a user study, and qualitative examples.

Weakness
- FaCT’s concepts are more consistent and interpretable than baselines but they are not necessarily align with human-defined categories. This at the end still limit how easily users can label the concepts.
- The metric C2-score relies on DINOV2, which could introduce errors if not reliable in all cases.
- It may need to test on more datasets to ensure it works broadly
- Lack of discussion on computational overhead

---

> ### Author Rebuttal · Authors · 2025-07-31
>
> We thank the reviewer Lkv1 for their detailed feedback. We are happy to see that the reviewer finds our approach and our evaluation novel and comprehensive. In the following, we will further address the Weaknesses **(W1-4)**, Questions **(Q1)** raised by the reviewer.
>
> **Human-alignment of Concepts (W1 & Q1):** There is an inherent difference between human-aligned models, such as CBMs, that are forced to learn a mapping to pre-defined (human) concepts and conventional models, that learn concepts that they deem useful, which do not need to correspond to what we as humans would like to see or even comprehend. The advantage of the latter is obvious in terms of performance, achieving much higher accuracy than e.g. CBMs at the cost of being less transparent. With FaCT, we aim at making the **concepts that the network learned** for decision-making transparent, which **do not need to align with pre-defined human concepts**. This allows us to maintain the performance of powerful models while allowing to interpret their reasoning. Here, a visualization of concepts will give a much better understanding of what is happening than text, given that we work on a vision task, which was also a key motivation for ProtoPNet, PipNet and others.
> Regardless, we can use off-the-shelf methods to label our concepts such as CLIP-Dissect [R1] which name individual neurons using a language-aligned model (i.e CLIP). Following your suggestion, we thus applied CLIP-Dissect [R1] to our SAE latents, and got the following names for the concepts (A-F) shown in Fig. 6 of the paper: (for each concept we show the top-3 matches):
>
>
> [A]: volleyball, rugby, balls
>
> [B]: jerseys, uniforms, players
>
> [C]: rugby, sergio, concentration
>
> [D]: gymnastics, acrobat, flexible
>
> [E]: volleyball, tournaments, serve
>
> [F]: basketball, layup, contested
>
> We find these names to be very consistent with how we would name the concepts in Fig. 6. We thank the reviewer for their suggestion and will happily add this discussion for the camera-ready.
>
>
>
> **Use of DiNOv2 for C2-Score (W2):** Indeed, output features of DiNOv2, or any other foundation model used within our C2-Score metric, should not be considered as the oracle or ground-truth semantic space. The underlying motivation of using DiNOv2 features is two fold: First, DiNOv2 has recently shown remarkable performance for similar tasks such as semantic correspondence [3, 52, 53], which makes it suitable for the context of evaluating consistency of concepts across images. Second, we intentionally steered away from evaluating the concepts with respect to human annotations, as the annotations are class-specific, subject to arbitrarily chosen granularity level, and are independent of what a model might learn to be useful for a decision.
> We have demonstrated this extensively in Fig C1 in  Appendix, where we show sample concepts that are not considered in Part-ImageNet annotations, while even the PCA visualization of DiNOv2 features seem to be better suited. We will make this comparison more prominent for the final version. Appreciating that DiNO is not the ground truth, we had complemented our analysis with a human user study to evaluate found concepts. We will further highlight this complementarity of evaluations and their individual limitations in the discussion.
>
> **Extending to More Datasets (W3):** We mainly experimented with ImageNet as it is a realistic dataset covering a wide range of different classes and diverse images, which many existing interpretability works such as ProtoPNet or PipNet fail to scale to. Following your suggestion, we evaluated FaCT also on the (uncropped) CUB dataset for ResNet-34 architecture. Similar to ImageNet experiments, we started with a pretrained B-cos checkpoint (ImageNet B-cos finetuned on CUB) and trained our SAEs at different layers, having the rest of the model frozen. The results can be seen in the table below.
>
> ### Table 3: Performance for ResNet34 on uncropped CUB dataset
> ### (Standard: 76.0 from [13])
> | Method | Accuracy | Consistency Gain (C2-Score) |
> | -  | -  | -  |
> | B-cos   | 78.8     | -   |
> | FaCT @Block 2/4     | 78.6     | 0.27 → 0.38   |
> | FaCT @Block 3/4     | 78.0     | 0.33 → 0.51   |
> | FaCT @Block 4/4     | 78.5     | 0.49 → 0.63   |
>
> We observe that FaCT works out of the box on smaller-scale datasets such as CUB and is able to maintain the accuracy (<0.8 % drop across), while training much faster than e.g. ProtoPNets. We will add the CUB results together with qualitative visualizations for the revision.
>
>
> **Discussing Computational Overhead (W4):** Following your suggestion, we measured the inference time over the entire ImageNet validation (50,000 Images). We did this for DenseNet-121 architecture on an A40 GPUs and averaged the results across three runs.
> ### Table 4: Time to process ImageNet’s Test set on DenseNet-121 (average across three runs)
> | Method            | Time (seconds) |
> | - | - |
> | B-cos    | 104 |
> | FaCT @Block 2/4  | 112  |
> | FaCT @Block 3/4  | 108  |
> | FaCT @Block 4/4  | 112  |
>
> We observe only a negligible increase in inference time, which is likely to reduce further with inference optimization (e.g. changing the SAE-hooks to fixed layers and using torch.compile). We will add these results for the final version.
>
> ---
> **[R1]:** CLIP-Dissect: Automatic Description of Neuron Representations in Deep Vision Networks, ICLR 2023

---

### Official Review · Reviewer_V8ET · 2025-07-04

**Clarity:** 2
**Significance:** 3
**Originality:** 3
**Rating:** 4
**Confidence:** 4

**Summary:**

The paper proposes FaCT, an inherently interpretable concept-based method that is capable of tracing the contributions that concepts, encoded throughout the network, have on the output produced by the model. The main motivation for this design lies on its goal of emphasizing the faithfulness that explanations produced from these concepts have on the underlying decision-making process followed by the model.

In practice, layer-level concept extraction is achieved through the use of sparse autoencoders (SAEs) and the links between the extracted concepts and the model outputs is enforced through a B-cos type architecture that is the basis of the proposed method. This is paired with a metric (C2-Score) to measure the consistency of extracted concepts across different examples.

Quantitative experiments as well as a user study conducted on the ImageNet-1k dataset show the capabilities of the proposed method in the downstream classification task, the faithfulness of the produced explanations, the intelligibility of the extracted concepts and their consistency.

**Questions:**

- When doing the, there is a very significant drop in performance in the performance went a relatively short amount of concepts is suppressed. While a drop is expected (assuming an explanation method is effective at assigning importance of concepts), the sharpness of the drop is rather abnormal and raises doubts. Is there any evidence or hypothesis of why this very sharp drop in performance is observed? How are the concepts been suppressed in that experiment? Is there any relation between the size (spatial extent) of concepts that are suppressed at earlier stages and the observed drop in performance?

- The third contribution and early parts of manuscript hint at the faithful visualization of concepts at the input-level, I would appreciate some clarification on what is meant by “faithful visualization”? Moreover, some clarify on how it differs from the aspects covered by the first two contributions could be helpful

- Does the user study  reported in Sec. 5.3 follow an existing protocol? are there steps taken to ensure not bias nor subjectivity affects the observed results?

    The second term of Eq. (14) covers a sum over H and W. This hints at a summation over a 2D matrix space, f(i) in this case.  Similarly in the first term the . operation takes f(I) as matrix. but according to l.141, isn’t f(I) an 1D vector of logits?

- When defining the random concept in Sec. 4 (l.180-183), what are the characteristics of its random attribution map in terms of its distribution, both across the spatial extent and intensity, that are considered for its generation? what is the effect that these two variables have in the effectiveness of the proposed C2-Score metric?

- What procedure is followed for computing the spatial extent/size of the concepts (l.210-215)? Is there a thresholding of an activation and/or attribution map taking place? if yes, how is that threshold defined?

- The ViT-S column in Fig.4 seems to suggest that the spatial extent of concepts is very similar at different depth, why is this the case? how is their size distributed at earlier layers?

**Ethical Concerns:**

["NO or VERY MINOR ethics concerns only"]

**Final Justification:**

Thanks for the extra efforts invested on the clarification of my question regarding the concept-deletion curves. I think the obtained insight of the 4 shared concepts is interesting and intriguing. If space allows, I would encourage the authors to include part of that finding in the main body of the paper.

Thanks for the clarification on faithful visualization, please add some related text in body of th paper in order to ensure this

Regarding the applicability of certain methods on specific metrics, I would argue that if the explanation provided by a method, e.g. the mentioned CRAFT, does not provide the means to assess its effect in the downstream prediction task, it might not be a valid competitor/baseline to compare against (as it is clearly impaired next to the proposed method). On the contrary, I would rather limit the comparison w.r.t. methods that allow for a complete comparison among different angles as defined by the considered metrics. In this regard

Regarding the evaluation of explanation methods via user studies. I agree with its usage as a means to complement a proper objective quantitative evaluation, and not as a the sole means for validation, especially if no proper protocol is followed. As that seems to be the same principle under which the user study presented in paper is conducted, the procedure described in the rebuttal seems to be more than sufficient. Having said for future (user-study) experiments/submissions, please have consider the line of work under the umbrella of Human-centered Explainable Artificial Intelligence, as that is research building on work from the Human-Computer Interaction community and have significant expertise conducting user studies.

Regarding concerns raised by other reviewers, considering the proposed method is bottom-up in nature, I do not find the lack of explicit human alignment as a strong weakness. I agree on that forcing alignment, as in classical CBMs, would steer the model and perhaps reduce its capabilities as a representation learning machine.
I agree with Reviewer qa8G on that one-stage training from scratch would be a very desirable capability to have. This would not only remove the dependency on pretrained B-cos models, but would in general remove the issues introduced by cascaded training schemes.

I thank the authors for their efforts in this rebuttal, in light of the above, I have opted to increase my initial rating.

**Limitations:**

In addition to the points listed as weaknesses I would stress the following aspects:

- The third contribution and early parts of manuscript hint at the faithful visualization of concepts at the input-level, this is an aspect that has not been properly defined nor evaluated in the rest of the manuscript


- In Sec. 5.1 (Fig.4, left), the performance of the proposed method in the downstream classification task is only reported with respect to the considered B-cos-variant of the considered architecture (i.e. ResNet-50, DenseNet-121, ViT-S) method and not the original vanilla architectures. Without that reference, it is hard to assess whether the proposed method has an effect on the performance achieved by the original architecture.

- The user study (Sec. 5.3) aims at evaluating the interpretabilty of a given concept by assessing whether users can retrieve meaning from visualizations. Here it is not clear how “meaning” is defined nor what protocol is followed to measure it.


- Due to the way the content is presented, the amount of conducted experiments and results reported; the level of detail provided in the content of the paper and depth of analysis of the results obtained in the conducted experiments is rather shallow. As indicated earlier, some parts of the manuscript are missing important details, other relevant parts of the manuscript have been delegated to the appendix. Perhaps a better structure must be adopted in order to ensure a proper depth is achieved in relevant areas.

**Paper Formatting Concerns:**

N>A>

**Quality:**

2

**Strengths And Weaknesses:**

**Strengths**

- The proposed method has been validated on top of a good variety of network architectures (CNNs and Tranformers) and, in some parts, compared w.r.t. various existing explanation methods.

- The proposed method seems to effectively ensure that the units used for producing the explanations, i.e. the concepts, do have a significant effect determining the outputs produced by the model.

- The explanations produced by the proposed method are extracted from different layers of the model, as such it enables nearly complete coverage of the architecture. This is in contrast to the common practice of existing efforts which produced explanations from features/concepts extracted from a very specific part/layer of the architecture.

- The validation of the proposed method includes both a quantitative, more machine-centered, evaluation as well as a user-study.


**Weaknesses**

- In the formal presentation of the proposed method, see Notation paragraph (l.112-115), the meaning of several symbols (with a wide established meaning) have been modified. This gets in the way when following the equations and other formal details that follow.

- Even when the evaluation of the proposed method included several existing methods from their literature, their inclusion in the different conducted experiments is not consistent.

- Some parts are missing important details, other relevant parts of the manuscript have been delegated to the appendix.

---

> ### Author Rebuttal · Authors · 2025-07-30
>
> We thank the reviewer V8ET for their constructive feedback. We are glad that they found the variety of our experiments and evaluation setup as a strength. In the following, we will further address the Weaknesses **(W1-3)**, Questions **(Q1-6)**, and Limitations **(L1-4)** raised.
>
> **Faithfulness of input-attributions (L1 & Q2):** One main limitation of existing concept-based methods (or similarly part-prototype networks) is that they struggle with correctly attributing the activation of a concept (or prototype) at late-layer to the image. These attributions are usually approximated with up-sampling similarity maps from the late-layer [10, 17, 46] or post-hoc attribution methods [1, 48] and have shown to be unfaithful visualizations [23, 32, 48]. In addition, such approximations are even worse in  ViT architectures, since the inductive (locality) bias of CNNs no longer holds. B-cos networks, however, provide model-inherent attributions that are faithful to the model in that **they are the actual model’s computation**. Precisely, for every input sample, they express the activation of the logit (or in our case concepts) as a linear mapping of the input, which summarizes the computation (See Eq. 3 for B-cos networks’s logits and Eq. 12 for FaCT’s concepts). This is independent of the layer of choice or the architecture. On top of the theoretical grounds, B-cos attributions have been shown to outperform other (approximate) attributions in terms of localization (see e.g. Fig. 6 in [9]).
>
> By using B-cos transforms for layers in FaCT, we are able to retain all of these benefits when it comes to visualizing concept attributions allowing us to precisely demonstrate where in the image the concept is being activated for. We would like to highlight this further by referring to Fig. E2 and E3 in Appendix E. Notice that it is only with such high-resolution faithful attributions at input-level that one can retrieve the correct meaning of each of the shown concepts, e.g. ‘Red Dots’.
>
> **Different baselines across experiments (W2):** Not all baselines make sense to compare in the context of every metric, as the metrics are task-specific. We elaborate below:
>
> ***a)** Accuracy (Fig 4):*  Post-hoc concept-based methods such as CRAFT do not directly use concepts for prediction, and hence one cannot measure “accuracy” for such methods. As a result, we compare against the B-cos model as a baseline, which is close to the accuracy of conventional models (see Table 1 below). For CRP the accuracy would correspond to the accuracy of the standard model, which as per your suggestion in (L2), we will add for easier reference.
>
> ***b)** Concept consistency (C2-score) (Fig 3):* We evaluate this against all baselines, i.e. against channels of the B-cos model, as well as post-hoc methods such as CRAFT and CRP.
>
> ***c)** Concept Deletion (Fig. 2):* This metric measures the accuracy of the importance value assigned to each concept, *given a fixed set of concepts*. Since each concept extraction method (FaCT, CRAFT, CRP, etc.) yields a distinct concept set, a direct comparison between them is not possible on this metric. As a result, we focus on comparing our approach of using B-cos attributions against alternative *concept attribution methods* used with these baselines, e.g. Sobol indices used in CRAFT and Saliency used in TCAV and VCC, and find that B-cos attributions achieve the sharpest drop, showing their effectiveness.
>
> ***d)** User Study (Fig. 5):* The goal of this experiment was to complement the C2-Score gains observed for SAE latents compared to B-cos channels, under the same visualizations, with the same architecture, weights, layer, with both having a shared concept basis. . In addition, since we were working with sampled concepts, we opted for having more samples rather than many methods with very few samples from each.
>
> **User Study (Q3 & L3):** To the best of our knowledge, there is no pre-defined protocol for evaluating fine-grained visual concepts. We have put our emphasis on developing a fair and unbiased study by introducing a double-blind control scenario, where we randomly distributed the users to 10 groups, each with randomly ordered samples from both methods. Similarly, when studying the effectiveness of visualizations, we performed counter-balanced AB/BA testing, where top-activating images of each concept were shown with and without the input-attribution in two different groups (see Fig. D2 in the Appendix). Regarding the definition of an interpretable concept with `meaining’, as this is abstractly defined in the literature, we followed the common approach of providing the user with examples. In particular, we provided examples of low-level, high-level, object-centric, and non-object-centric concepts to the users (see Fig. D1).
>
> **Vanilla Baseline (L2):** We will add the standard baselines for the final version for easier comparison. For convenience, we have put a comparison table below, where we compare FaCT, B-cos, and the vanilla architecture.
> ### Table 1: Comparison to Vanilla Baseline
> | Architecture | Vanilla Accuracy | B-cos Accuracy |  (FaCT@LastBlock) Accuracy  | Consistency Gain |
> |-|-|-|-|-|
> | ResNet50 | 80.9 | 79.5 | 77.3 | 0.38 → 0.49 |
> | DenseNet121 | 74.4* | 77.3 | 75.2 | 0.31 → 0.50 |
> | ViT_c-S | 75.8 | 74.5 | 73.2 | 0.35 → 0.51|
>
> \* Trained with torchvision’s v1 recipe, as v2 is not available for DenseNets. (Other rows are trained with the v2 recipe.)
>
> We observe a significant increase in the consistency (i.e C2-Score) of concepts, while remaining on-par on ImageNet accuracy across architectures.
>
> **Concept-Deletion Curves (Q1):**  We were indeed also curious about such a sharp drop for early layers (Block 2/4) seen in Fig. 2, as well as Fig. F1 in Appendix. For these early-layer experiments, we investigated the first few concepts that are removed, and found a very small subset of latents that occur on every image. This was primarily the case for early-layer experiments. In the case of Fig. 2, there are exactly 4 concepts that occur on 100% of the samples. We had also identified these while training the SAEs, discussed it in Appendix A (lines 675-682), and found similar phenomena in the existing literature (see points with frequency of 1 in Figure 3 of [31]). We hypothesize that these are mean feature vectors that are used for reconstruction in every sample. It is the removal of these latents that results in the sharp drops for early-blocks in Fig. 2. Nevertheless, as we use the same concept-set for evaluating the concept-importance methods, this still shows that B-cos is best in identifying the most impactful concepts.
>
> To investigate the identification of most relevant concepts beyond these “always-on” shortcut concepts, we now investigated concept deletion when not allowing removal of these four latents and evaluated on the rest of the concept. We plotted the results similar to Fig.2 and found B-cos contributions to again consistently out-perform other attribution methods on concepts. Since we cannot upload images for the rebuttal, we report the Area Under the Deletion-Curve here in Table 2: (Lower means faster drop, hence better).
>
> ### Table 2: Area under Deletion Curve for Block 2/4 on DenseNet-121
> | Method | Removing the 4 concepts is allowed (Fig. 2) | Removing the 4 concepts is not allowed |
> |-|-|-|
> | B-cos  | 0.9 | 24.0 |
> | Saliency | 56.7 | 73.7 |
> | Sobol (n=4) | 108.2 | 111.0 |
> | Random | 98.9 | 116.2 |
>
> We observe that the sharp drop disappears once we do not allow the removal of these four concepts for any method, yet we observe that B-cos contributions still significantly outperform other concept-importance measures. We thank the reviewer for raising this point. We will make the discussion around always-occurring concepts at early layers (lines 675-682 in Appendix A) more prominent for the final version, while also adding this new experiment which excludes removal of these concepts.
>
> **Notation in Eq. 14 (Q3.5):** Thanks for pointing this out. Indeed, in Eq (14), $f(I)$ should be replaced by $H(I)$, which refers to the foundation model’s features. We will correct this for the revision.
>
> **Random Concept in C2-Score (Q4):** The random concept activates on every image and assigns random importance to each pixel (sampled uniformly from  within [0, 1]). In a multi-class setup, the consistency (Eq. 16) of random concept was at 0.004 ($\sigma$ 0.0005), averaged across three seeds, while on a per-class setup it was 0.13 ($\sigma$ 0.021), averaged across the 1000 classes. This agrees with the original assumption that when the evaluation set $\mathcal{D}$ is restricted to a single category, all the output features of DiNOv2 become more similar. This motivated us to define C2-score as the difference w.r.t the random baseline.
>
> **Spatial Extent (Q5):** The spatial extent is the average number of pixels required to cover 80% of the total positive attribution of a concept. This was defined so that we can demonstrate the variety of concepts, as opposed to prior work that often assumes concepts to fit in small patches [17, 33]. We have defined the spatial extent in Appendix E (Lines 786-792) and will make sure to make it more prominent in the revision.
>
> **Spatial Extent of ViTs (Q6):** We believe the similarity in spatial extent across two layers is due to the full-image receptive field at all the layers of ViT. Following your suggestion, we will run an experiment at a very early-layer and extend the plot for the revision.
>
> **Clarity and Notation (W1, W3, L4):** We will take all of the questions raised by the reviewer into account and revise for clarity. Regarding the notation, while we were trying to closely follow the notation in B-cos paper [9], we will make sure to adjust it to be more intuitive. We will also highlight references for the Appendix sections more clearly. We would be grateful if the reviewer could specifically mention what they find missing so that we revise for the revision.

---

> > ### Comment · Reviewer_V8ET · 2025-08-05
> > **RE: Rebuttal**
> >
> > Thanks for the extra efforts invested on the clarification of my question regarding the concept-deletion curves. I think the obtained insight of the 4 shared concepts is interesting and intriguing. If space allows, I would encourage the authors to include part of that finding in the main body of the paper.
> >
> > Thanks for the clarification on faithful visualization, please add some related text in body of the paper.
> >
> > Regarding the applicability of certain methods on specific metrics, I would argue that if the explanation provided by a method, e.g. the mentioned CRAFT, does not provide the means to assess its effect in the downstream prediction task, it might not be a valid competitor/baseline to compare against (as it is clearly impaired next to the proposed method). On the contrary, I would rather limit the comparison w.r.t. methods that allow for a complete comparison among different angles as defined by the considered metrics. In this regard
> >
> > Regarding the evaluation of explanation methods via user studies. I agree with its usage as a means to complement a proper objective quantitative evaluation, and not as a the sole means for validation, especially if no proper protocol is followed. As that seems to be the same principle under which the user study presented in paper is conducted, the procedure described in the rebuttal seems to be more than sufficient. Having said for future (user-study) experiments/submissions, please have consider the line of work under the umbrella of Human-centered Explainable Artificial Intelligence, as that is research building on work from the Human-Computer Interaction community and have significant expertise conducting user studies.
> >
> > Regarding concerns raised by other reviewers, considering the proposed method is bottom-up in nature, I do not find the lack of explicit human alignment as a strong weakness. I agree on that forcing alignment, as in classical CBMs, would steer the model and perhaps reduce its capabilities as a representation learning machine.
> > I agree with Reviewer qa8G on that one-stage training from scratch would be a very desirable capability to have. This would not only remove the dependency on pretrained B-cos models, but would in general remove the issues introduced by cascaded training schemes.
> >
> > I thank the authors for their efforts in this rebuttal, in light of the above, I have opted to increase my initial rating.

---

### Comment · Area_Chair_hx1Q · 2025-08-04
**Author-Reviewer Discussions**

Dear Reviewers,

Thanks for taking the time to review for NeurIPS. For those who haven’t responded yet, please do the following ASAP: carefully read the other reviews and the authors responses, and reply to the authors regarding whether your questions or concerns have been addressed. If any issues remain unresolved, please clearly specify them so that the authors have a fair opportunity to respond before the author-reviewer discussion period ends in fewer than three days (by August 6, 11:59pm AoE).

Thank you again for your contribution to the review process.

Best,
AC

---

### Author Response · Authors · 2025-08-08
**Appreciation for Reviewers’ Insights and Support**

We are thankful for the high-quality and constructive reviews from all of the four reviewers, and are happy to see they find our proposed approach novel and a valuable contribution to the community. We will ensure to include all of the discussions from the rebuttal in the new draft.

We are especially thankful to reviewers V8ET and qa8G for staying so engaged during the discussion.

---

### Decision · Program_Chairs · 2025-09-17

**Decision:**

Accept (poster)

**Comment:**

This paper introduces FaCT, an inherently interpretable concept-based method for explaining neural network decisions. Unlike previous methods, FaCT breaks down predictions into clear and faithful concepts across all layers of the network. The proposed model integrates sparse autoencoders and B-cos networks to ensure these concepts truly reflect the model’s reasoning. The authors also propose a new metric, C^2-score, to measure how consistent concepts are across images. They validated their approach using quantitative experiments on ImageNet, qualitative visualizations, and a user study.

Reviewers praised the clear ideas, strong experiments, and good writing of the paper. The method is applicable to different types of networks and maintains competitive accuracy while improving interpretability. Some concerns were raised by reviewers in the initial reviews, like how easy it is to label concepts, limited dataset evaluation, and comparisons with other baselines, but the authors addressed most concerns well. All reviewers have positive ratings for this paper. Overall, this paper is an important step forward in making deep learning more explainable, and is recommended for acceptance.